# TransMLA: Migrating GQA Models to MLA with Full DeepSeek Compatibility and Speedup

**Fanxu Meng**[1,*] **Pingzhi Tang**[1,*] **Zengwei Yao**[4], **Xing Sun**[3], **Muhan Zhang**[1,2,†]
[1]Institute for Artificial Intelligence, Peking University
[2]State Key Laboratory of General Artificial Intelligence, BIGAI
[3]Tencent Youtu Lab, Shanghai, China
[4]Xiaomi Corp., Beijing, China
https://github.com/MuLabPKU/TransMLA

## Abstract

Modern large-language models often face communication bottlenecks on current hardware rather than computational limitations. *Multi-head latent attention* (*MLA*) addresses this by compressing the key-value cache using low-rank matrices, while the Absorb operation prevents the KV cache from reverting to its original size, significantly boosting both training and inference speed. Despite the success of DeepSeek V2/V3/R1, most model providers have heavily invested in optimizing GQA-based models and, therefore, lack strong incentives to retrain MLA-based models from scratch. This paper demonstrates that MLA provides superior expressive power compared to GQA with the same KV cache overhead, thereby offering a rationale for transitioning from GQA to MLA. In addition, we introduce TransMLA, a framework that seamlessly converts any GQA-based pre-trained model (e.g., LLaMA, Qwen, Gemma, Mistral/Mixtral) into an MLA-based model. For the first time, our method enables *direct conversion of these models into a format compatible with DeepSeek's codebase*, allowing them to fully leverage the existing, highly-optimized support for the DeepSeek architecture within inference engines like vLLM and SGlang. By compressing 93% of the KV cache in LLaMA-2-7B, we achieve a **10x speedup** with an 8K context length while maintaining meaningful output. Moreover, the model requires only **6B tokens** for fine-tuning to recover comparable performance across multiple benchmarks. TransMLA provides a practical path for migrating GQA-based models to the MLA structure, and when combined with DeepSeek's advanced optimizations—such as FP8 quantization and Multi-Token Prediction—further inference acceleration can be achieved.

## 1 Introduction

Advanced Large language models (LLMs)—such as GPT-4o [1], Claude 3.7 Sonnet [2], Gemini-2.5 [3], LLaMA-4 [4], Mistral-3 [5], Qwen-3 [6], DeepSeek V3/R1 [7, 8], Gemma-3 [9], and Phi-4 [10]—represent a rapidly evolving frontier for both research and applications. At their core, LLMs rely on next-token prediction [11, 12]: tokens are generated sequentially, a process that requires self-attention to be computed over all preceding tokens at each generation step. To avoid the computationally expensive recalculation of these states, implementations store the intermediate key–value (KV) pairs in a cache. Yet, as model and context sizes grow, the KV cache itself becomes a major bottleneck for inference.

---

*Equal contribution.
†Corresponding author: muhan@pku.edu.cn

39th Conference on Neural Information Processing Systems (NeurIPS 2025).

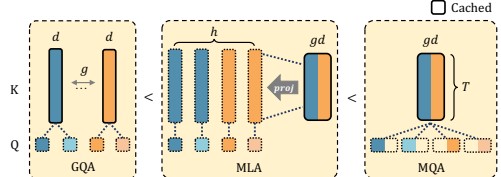
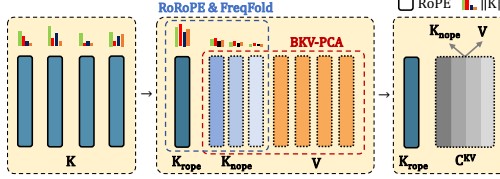

(a) Given the same KV cache size, the expressiveness increases in the order of GQA, MLA, and MQA.

(b) TransMLA concentrates positional information into $K_{\text{rope}}$ and compresses $K_{\text{nope}}$ and $V$.

Figure 1: (a) GQA, MLA, and MQA can be equivalently transformed in one direction, illustrating a gradual increase in expressive power. (b) RoRoPE aggregates positional information in the first head, eliminating the need for RoPE in others. FreqFold further enhances this effect. After balancing the magnitudes of $K_{\text{rope}}$ and $V$, a joint low-rank approximation is applied to compress the KV cache.

To mitigate these challenges, Group-Query Attention (GQA) [13] partitions the query heads into groups, where every head within a group shares a single key and value head. GQA thus generalizes both Multi-Query Attention (MQA) [14] and Multi-Head Attention (MHA) [15]: it degenerates to MQA when a single group is used, and reduces to MHA when the number of groups equals the number of query heads. While both GQA and MQA cut the size of the KV cache relative to MHA, they do so at the cost of model quality. Post-training KV-cache compression techniques—such as Duo-Attention [16], KiVi [17], KV-Quant [18], $H_2O$ [19] and Palu [20]—further shrink memory usage, but their non-standard implementations demand specialized optimizations, hindering widespread adoption.

Multi-Head Latent Attention (MLA)—introduced with DeepSeek V2 [21] and further refined in DeepSeek-V3 [7]/-R1 [8]—offers a pre-trained KV-cache compression strategy that strikes an excellent balance between computational efficiency and model quality. Models equipped with MLA deliver state-of-the-art results while driving training and inference costs to new lows. Moreover, the DeepSeek team's ongoing commitment to open-source releases provides highly optimized implementations and deployment recipes, making these advances readily accessible to the community.

In this paper, we first prove that MLA consistently offers *higher expressive power* than GQA under the same KV cache overhead, which theoretically explains the advantage of MLA. However, a practical hurdle preventing model vendors from switching to MLA is the substantial prior investment on GQA-based models. This motivates us to ask: can we *seamlessly convert a GQA-based pretrained model*, such as LLaMA [4] and Qwen [6], *to MLA so that we can inherit the model weights and pretraining effort*, rather than training MLA from scratch?

A key obstacle to converting a GQA-based model to MLA is that the *Absorb* operation [20], which DeepSeek uses to switch between the compute and memory-efficient modes, is blocked. This occurs because every query / key head has its own Rotary Positional Embedding (RoPE) [22]. Borrowing from DeepSeek's *Decoupled RoPE* scheme, we concentrate the positional signal in $K$ into a small subset of dimensions, $K_{\text{rope}}$. The remaining dimensions, $K_{\text{nope}}$, contain negligible positional content. This decoupling allows us to drop the RoPE from $K_{\text{nope}}$ and then absorb its up-projection into the query projection—mirroring the DeepSeek methodology—thereby enabling seamless MLA conversion.

To efficiently concentrate positional information into fewer dimensions, we introduce **RoRoPE**—a novel technique that performs principal component analysis (PCA) on the key output, applies rotation across the two ends of RoPE, and consolidates the principal components of all attention heads into the dimensions of the first attention head. We theoretically prove that the product remains invariant after rotating the query and key using an orthogonal matrix $\mathbf{U}$, as long as $\mathbf{U}$ satisfies two conditions: *(1) rotation occurs only within the same dimensions across all attention heads and (2) the real and imaginary components of RoPE are rotated in the same manner.* Additionally, by exploiting the frequency similarity between adjacent RoPE dimensions, we propose **FreqFold**, a technique that improves the concentration efficiency.

Finally, we find that the $\ell_2$-norm of $K_{\text{nope}}$ is much larger than that of $V$. If we run PCA on the concatenated matrix $\begin{bmatrix} K_{\text{nope}}; V \end{bmatrix}$ without adjustment, the principal components are dominated by $K_{\text{nope}}$, leading to a severe loss of information from the value subspace and a sharp decrease in accuracy. We therefore introduce a Balanced Key–Value (BKV) procedure: we first rescale $K_{\text{nope}}$ and

$V$ so that their norms match, and only then perform joint PCA. This simple normalization restores the balance between the two subspaces and delivers a marked improvement in compression quality.

The above innovations collectively form our TransMLA method. Using TransMLA, we compressed the KV cache of LLaMA-2 by 68.75%, with only a 1.65% performance drop across 6 benchmarks for training free. In contrast, a concurrent method, MHA2MLA [23], experienced a 21.85% performance decline. At a compression rate of 93%, the model still maintained meaningful responses, and after training with 6B tokens, its performance was mostly restored. We evaluated both the original (GQA/MHA) models and their TransMLA-converted counterparts on different hardware setups using vLLM. The TransMLA models achieved up to a 10x speedup over the originals, demonstrating the significant potential of our approach. Moreover, the TransMLA models are fully compatible with DeepSeek's code, enjoying DeepSeek's ecosystem to accelerate inference and seamlessly integrate with various hardware and frameworks.

## 2 Related Work

Autoregressive decoding in large language models necessitates storing past activations—the key–value (KV) pairs—in a cache to avoid recomputation. Because the size of this cache grows linearly with sequence length, its memory footprint quickly becomes the limiting factor for very long contexts. Consequently, shrinking the KV cache without compromising accuracy has become a pivotal research focus, motivating a spectrum of architectural innovations and compression strategies.

Multi-Query Attention (MQA) [14] and Group-Query Attention (GQA) [13] shrink the KV cache by letting every query in a group share a single key and value head. Although both schemes save memory relative to Multi-Head Attention (MHA) [15], they usually give up some accuracy. Multi-Head Latent Attention (MLA)—introduced with DeepSeek V2 [21] and refined in later releases DeepSeek V3/R1 [21, 24]—offers a more favorable trade-off, delivering near-state-of-the-art quality while cutting training and inference costs. Grouped Latent Attention (GLA) [25] provides a parallel-friendly implementation of latent attention that further accelerates MLA inference. By contrast, Tensor Product Attention (TPA) [26] tackles the memory bottleneck by dynamically factorizing activations, slashing the runtime KV cache by an order of magnitude, but it necessitates training the model from scratch. TransMLA fills this gap: rather than proposing yet another attention variant, it converts an existing GQA model into an MLA model with only light fine-tuning, restoring accuracy while inheriting MLA's memory and speed advantages.

Another approach is to optimize the KV cache of existing pre-trained models. For example, dynamic token pruning is employed by LazyLLM [27], A2SF [28], and SnapKV [29]. These methods selectively prune less important tokens from the KV cache. Sharing KV representations across layers, as in YONO [30], MiniCache [31], and MLKV [32], reduces memory by reusing the same KV cache across multiple layers. This can drastically lower memory usage and speed up inference. Although effective, both families of methods usually require custom kernels or runtime tweaks, complicating deployment and limiting adoption. TransMLA, by contrast, plugs directly into the mature DeepSeek ecosystem—converted checkpoints load out-of-the-box, delivering MLA-level speed-ups across every DeepSeek-supported platform.

There are two works most related to TransMLA. One is Palu [20], which reduces KV cache size by applying low-rank decomposition on both the keys and values, enabling speedup through tailored optimizations. However, Palu does not specifically handle RoPE, which prevents it from using the Absorb operation during inference. Therefore, Palu needs to project the compressed representations back to their original size. This projection incurs significant computational overhead during inference, limiting the overall acceleration. Another concurrent work, MHA2MLA [23], also claims to convert MHA to MLA and decouple RoPE from the main computational path. It is important to clarify that TransMLA is not simply a GQA extension of MHA2MLA—both TransMLA and MHA2MLA support MHA and GQA architectures. However, MHA2MLA determines which RoPE dimensions to remove solely based on the norms of the query and key vectors, which tends to cause larger information loss when pruning the same proportion of positions. Also, the distribution of important dimensions in MHA2MLA is uneven, requiring sparse indexing that complicates optimization and acceleration. Their work reports compression ratios of the KV cache but does not demonstrate actual inference speedup. Furthermore, MHA2MLA directly applies joint singular value decomposition to KV, resulting in higher loss compared to our balanced key-value PCA method.

# 3 Preliminary

**Rotary Position Embedding and Group Query Attention**  Details are provided in Appendix A.

**Multi-Head Latent Attention**  MLA operates on an input sequence, where $\mathbf{x}_t \in \mathbb{R}^D$ is the embedding of the $t$-th token and $D$ is the model's hidden dimension. To save KV cache, it first computes low-rank latent features $\mathbf{c}_t^{KV}$ using a down-projection matrix $W^{DKV} \in \mathbb{R}^{r_{kv} \times D}$. Then, up-projection matrices $W^{UK} \in \mathbb{R}^{hd \times r_{kv}}$ and $W^{UV} \in \mathbb{R}^{hd \times r_{kv}}$ are used to derive the key ($\mathbf{k}$) and value ($\mathbf{v}$) representations for all $h$ heads, where $d$ is the per-head dimension. Similarly, MLA decomposes the query projection (using $W^{DQ} \in \mathbb{R}^{r_q \times D}$ and $W^{UQ} \in \mathbb{R}^{hd \times r_q}$, where $r_q$ is the query rank) to reduce activation memory. For positional embedding, MLA uses a decoupled RoPE strategy. This employs additional multi-head queries $\mathbf{q}_{t,i}^R \in \mathbb{R}^{d^R}$ and a shared key $\mathbf{k}_t^R \in \mathbb{R}^{d^R}$ to carry the RoPE signal, which are generated from $W^{QR} \in \mathbb{R}^{hd^R \times r_q}$ and $W^{KR} \in \mathbb{R}^{d^R \times D}$ ($d^R$ is the per-head RoPE dimension). The computation of the key and query representations is formulated as follows:

$$\mathbf{c}_t^{KV} = W^{DKV}\mathbf{x}_t, \qquad\qquad\qquad \mathbf{c}_t^Q = W^{DQ}\mathbf{x}_t,$$

$$[\mathbf{k}_{t,1}^C; \mathbf{k}_{t,2}^C; ...; \mathbf{k}_{t,h}^C] = \mathbf{k}_t^C = W^{UK}\mathbf{c}_t^{KV}, \qquad [\mathbf{q}_{t,1}^C; \mathbf{q}_{t,2}^C; ...; \mathbf{q}_{t,h}^C] = \mathbf{q}_t^C = W^{UQ}\mathbf{c}_t^Q,$$

$$\mathbf{k}_t^R = \text{RoPE}_t(W^{KR}\mathbf{x}_t), \qquad [\mathbf{q}_{t,1}^R; \mathbf{q}_{t,2}^R; ...; \mathbf{q}_{t,h}^R] = \mathbf{q}_t^R = \text{RoPE}_t(W^{QR}\mathbf{c}_t^Q),$$

$$\mathbf{k}_{t,i} = [\mathbf{k}_{t,i}^C; \mathbf{k}_t^R], \qquad (1) \qquad\qquad \mathbf{q}_{t,i} = [\mathbf{q}_{t,i}^C; \mathbf{q}_{t,i}^R]. \qquad (2)$$

MLA features a dual-mode capability, tailored for different operational stages. For the compute-intensive training phase, it adopts a configuration resembling standard MHA (Equation (3)), offering slightly lower computational overhead than conventional MHA. Conversely, for communication-intensive inference, it can seamlessly switch to an MQA-like setting (Equation (4)). In this latter mode, the latent features function as a shared large KV head, interacting with all query and output heads to efficiently produce the final output. This mechanism, known as the Absorb operation, is crucial for accelerating inference.

$$[\mathbf{v}_{t,1}^C; \mathbf{v}_{t,2}^C; ...; \mathbf{v}_{t,h}^C] = \mathbf{v}_t^C = W^{UV}\mathbf{c}_t^{KV}, \qquad \hat{\mathbf{q}}_{t,i} = [W_i^{UK\top}\mathbf{q}_{t,i}^C; \mathbf{q}_{t,i}^R], \quad \hat{\mathbf{k}}_t = [\mathbf{c}_t^{KV}; \mathbf{k}_t^R],$$

$$\mathbf{o}_{t,i} = \sum_{j=1}^t \text{softmax}_j\Big(\frac{\mathbf{q}_{t,i}^T\mathbf{k}_{j,i}}{\sqrt{d+d^R}}\Big)\mathbf{v}_{j,i}^C, \qquad \hat{\mathbf{o}}_{t,i} = \sum_{j=1}^t \text{softmax}_j\Big(\frac{\hat{\mathbf{q}}_{t,i}^T\hat{\mathbf{k}}_j}{\sqrt{d+d^R}}\Big)\mathbf{c}_j^{KV},$$

$$\mathbf{y}_t = W^O[\mathbf{o}_{t,1}; \mathbf{o}_{t,2}; ...; \mathbf{o}_{t,h}], \qquad (3) \qquad \mathbf{y}_t = W^O[W_1^{UV}\hat{\mathbf{o}}_{t,1}; ...; W_h^{UV}\hat{\mathbf{o}}_{t,h}], \qquad (4)$$

where $W_i^{\{UK,UV\}}$ denotes slices of the projection matrices corresponding to the $i$-th attention head.

One of the main contributions of this paper is the seamless support for the Absorb operation, significantly enhancing inference speed.

# 4 TransMLA

In this section we formally present TransMLA, motivated by two observations:

**1. For a fixed KV-cache budget, MLA is strictly more expressive than GQA.** As proven in Appendix B and illustrated in Figure 1a, any GQA layer can be rewritten as an MLA layer by introducing a single additional projection matrix. The reverse transformation is not always possible, implying that MLA subsumes GQA. This equivalence holds even when RoPE are present, although it requires the equivalent MLA layer to be expressed in the *absorbed* form.

**2. Inference acceleration occurs when MLA uses a smaller KV cache.** Although one can build an MLA-equivalent representation of a GQA model, speedups arise only if the number of stored KV vectors is actually reduced. TransMLA achieves this by converting the GQA-based network into a DeepSeek-like MLA architecture while compressing its KV cache. This allows the transformed model to run directly on DeepSeek's inference stack and realize the full memory–latency benefits.

## 4.1 Merging Grouped Heads to a Latent Head

The core insight behind establishing the strictly greater expressive power of MLA over GQA, and the first step in the TransMLA transformation, lies in unifying GQA's $g$ grouped key-value (KV)

heads into a single MLA latent representation. This transformation replicates GQA's `repeat_kv` mapping, where each query $i$ maps to a specific KV group $j = \lceil i/(h/g) \rceil$. To achieve this, we initialize the MLA up-projection $W_i^{UK}, W_i^{UV}$ as sparse block matrices: for each $i$-th query head, only the block corresponding to its designated $j$-th KV group is set to an identity matrix, while all others are zero. This ensures that a single key representation can be shared across $g$ query heads while maintaining interactions solely with its designated partner. $W_i^{UV}$ is initialized analogously to map to the $j$-th value head. Consequently, the $g$ identical RoPE rotations must also be consolidated into a single operator, denoted $\overline{\text{RoPE}}$, which applies the original rotation pattern (of per-head dimension $d$) repeatedly every $d$ dimensions to the unified key. In this way, the computation of GQA attention is transformed into the following form:

$$[\mathbf{q}_{t,1}; \mathbf{q}_{t,2}; \dots; \mathbf{q}_{t,h}] = \mathbf{q}_t = W^Q \mathbf{x}_t, \quad [\mathbf{c}_t^K; \mathbf{c}_t^V] = \mathbf{c}_t^{KV} = W^{DKV} \mathbf{x}_t, \tag{5}$$

$$\hat{\mathbf{q}}_{t,i}^R = \overline{\text{RoPE}}_t \left( W_i^{UK^\top} \mathbf{q}_{t,i} \right), \quad \hat{\mathbf{k}}_t^R = \overline{\text{RoPE}}_t \left( \mathbf{c}_t^K \right), \quad \hat{\mathbf{v}}_t = \mathbf{c}_t^V, \tag{6}$$

$$\hat{\mathbf{o}}_{t,i} = \sum_{j=1}^{t} \text{softmax}_j \left( \frac{\hat{\mathbf{q}}_{t,i}^{R^\top} \hat{\mathbf{k}}_j^R}{\sqrt{d}} \right) \hat{\mathbf{v}}_j, \quad \mathbf{y}_t = W^O [W_1^{UV} \hat{\mathbf{o}}_{t,1}; \dots; W_h^{UV} \hat{\mathbf{o}}_{t,h}], \tag{7}$$

where:

$$W^{DKV} = [W^K; W^V] \in \mathbb{R}^{2gd \times D},$$

$$W_i^{UK} = W_i^{UV} \in \mathbb{R}^{d \times gd}, \quad W_i^{UK}[k,l] = \begin{cases} 1 & \text{if } l = (j-1)d + k, \\ 0 & \text{otherwise.} \end{cases}$$

This GQA-equivalent formulation (Equation (6)-(7)) is crucial because it makes the model structurally compatible with the `absorb` operation, enabling seamless switching between execution modes. However, this equivalence alone provides no acceleration. It is evident that the total KV cache size remains unchanged ($\mathbf{c}_t^{KV} \in \mathbb{R}^{2gd}$), which is identical to the original GQA model. Furthermore, as shown in Equation (6), the effective attention dimension increases from $d$ to $gd$ (since $\hat{\mathbf{q}}_{t,i}^R$ and $\hat{\mathbf{k}}_t^R$ are both in $\mathbb{R}^{gd}$), leading to higher computational costs. To achieve actual acceleration, compressing the KV cache is therefore essential. This unified representation is advantageous for compression, as merging multiple KV heads allows for better identification of shared principal components. Moreover, this merged-key structure is a prerequisite for efficiently decoupling RoPE in the subsequent steps.

Overview of the RoRoPE decoupling pipeline. Blue lines and orange lines denote the real and imaginary components, respectively. We first gather dimensions with the same rotational frequency ($\theta_l$) across all $g$ heads. A joint Principal Component Analysis (PCA) is then applied to these $g$-dimensional real and imaginary components. This process concentrates the dominant positional signal into a single principal component for each frequency, which is then represented by a standard RoPE (RoPE 1, RoPE 2), while the remaining components are decoupled (NoPE 3-8).

### 4.2 Head-wise Rotation for Decoupled RoPE with Minimal Loss

To enable the essential KV cache compression identified in the previous section, we introduce a method, **RoRoPE**. As illustrated in Figure 2, applying RoRoPE to the merged head removes the bulk of the positional signal from K.

In the GQA-equivalent form (Equation (7)), the attention dot product $\hat{\mathbf{q}}_{t,i}^{R^\top} \hat{\mathbf{k}}_j^R$ is computed over the merged $gd$-dimensional vectors. Recall that RoPE operates by rotating $d/2$ independent 2D subspaces. Because our $\overline{\text{RoPE}}$ operator applies the same rotation pattern to all $g$ heads, we can regroup this total dot product by summing over these $d/2$ independent subspaces. We define the *real* and *imaginary* components (Appendix A) of the $l$-th 2D subspace (where $l \in [1, d/2]$) as the $(2l-1)$-th and $(2l)$-th dimensions of any single head, respectively. Let $\hat{\mathbf{q}}_{t,i}^{(l,\text{real})}$ and $\hat{\mathbf{q}}_{t,i}^{(l,\text{imag})}$ be the $g$-dimensional vectors formed by gathering all real (dims $[2l-1, 2l-1+d, \dots]$) and imaginary (dims $[2l, 2l+d, \dots]$) components, respectively, from the $g$ (absorbed) query heads. The key vectors $\hat{\mathbf{k}}_j^{(l,\text{real})}$ and $\hat{\mathbf{k}}_j^{(l,\text{imag})}$ are defined similarly from $\hat{\mathbf{k}}_j^R$. The total $gd$-dimensional dot product can then be expressed as the sum of $d/2$ independent $2g$-dimensional dot products:

$$\hat{\mathbf{q}}_{t,i}^{R^\top} \hat{\mathbf{k}}_j^R = \sum_{l=1}^{d/2} \overline{\text{RoPE}}_{t,l} \left( \left[ \hat{\mathbf{q}}_{t,i}^{(l,\text{real})}; \hat{\mathbf{q}}_{t,i}^{(l,\text{imag})} \right] \right)^\top \overline{\text{RoPE}}_{j,l} \left( \left[ \hat{\mathbf{k}}_j^{(l,\text{real})}; \hat{\mathbf{k}}_j^{(l,\text{imag})} \right] \right), \tag{8}$$

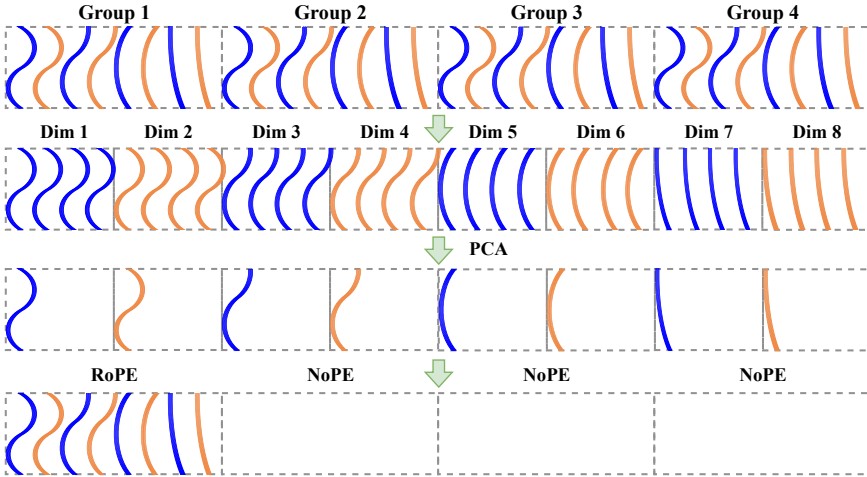

Figure 2: Overview of the RoRoPE decoupling pipeline. Blue lines and orange lines denote the real and imaginary components, respectively. We first gather dimensions with the same rotational frequency ($\theta_l$) across all $g$ heads. A joint Principal Component Analysis (PCA) is then applied to these $g$-dimensional real and imaginary components. This process concentrates the dominant positional signal into a single principal component for each frequency, which is then represented by a standard RoPE

where

$$\overline{\text{RoPE}}_{t,l}\left(\left[\mathbf{x}_t^{(l,\text{real})}; \mathbf{x}_t^{(l,\text{imag})}\right]\right) = \cos t\theta_l \left[\mathbf{x}_t^{(l,\text{real})}; \mathbf{x}_t^{(l,\text{imag})}\right] + \sin t\theta_l \left[-\mathbf{x}_t^{(l,\text{imag})}; \mathbf{x}_t^{(l,\text{real})}\right]. \quad (9)$$

When the $l$-th concatenated real and imaginary components of $\hat{\mathbf{q}}_{t,i}$ and $\hat{\mathbf{k}}_j$ are multiplied by an orthogonal matrix $\mathbf{U}_l \in \mathbb{R}^{g \times g}$, the inner product with RoPE applied remains invariant. Specifically,

$$\sum_{l=1}^{d/2} \overline{\text{RoPE}}_{t,l}\left(\left[\mathbf{U}_l\hat{\mathbf{q}}_{t,i}^{(l,\text{real})}; \mathbf{U}_l\hat{\mathbf{q}}_{t,i}^{(l,\text{imag})}\right]\right)^{\top} \overline{\text{RoPE}}_{j,l}\left(\left[\mathbf{U}_l\hat{\mathbf{k}}_j^{(l,\text{real})}; \mathbf{U}_l\hat{\mathbf{k}}_j^{(l,\text{imag})}\right]\right) = \hat{\mathbf{q}}_{t,i}^{R\top}\hat{\mathbf{k}}_j^R. \quad (10)$$

This invariance holds because $\overline{\text{RoPE}}$ is a linear rotation independently applied to different heads, and any orthogonal transformation applied to the $g$-dimensional components preserves the inner product. However, this reveals a critical constraint: the same orthogonal matrix $\mathbf{U}_l$ must be applied to both the real and imaginary components within each 2D subspace. For a detailed proof and our proposed solution, please refer to Appendix C.

In practice, our **RoRoPE** method uses Principal Component Analysis (PCA) to compute the required orthogonal matrices $\{\mathbf{U}_l\}_{l \in \{1,\ldots,d/2\}}$ from the key activations for each $l$-th subspace. To apply these rotations (as required by Equation (10)), we transform the weight matrices $W^K$ and $W^{UK}$ directly. This is because rotating the activations ($\hat{\mathbf{q}}_{t,i}, \hat{\mathbf{k}}_j$ is equivalent to rotating their respective generation matrices. The rotation effectively concentrates the essential information into the first few components (heads), and in turn, allows us to perform the final decoupling: we remove the RoPE encoding from the non-principal components, preserving positional information only within the principal components.

However, applying PCA directly to each $2g$-dimensional subspace independently is suboptimal, as this space is often too small for effective compression. We address this by exploiting a key property of RoPE: adjacent rotational frequencies are often similar ($\theta_l \approx \theta_{l+1}$). This similarity allows us to treat adjacent subspaces as effectively equivalent, grouping them to perform PCA over a larger, combined latent space. This technique, which we call **FreqFold**, achieves superior compression by identifying principal components across a broader set of dimensions. We defer a detailed analysis of FreqFold to Appendix D.

### 4.3 Balancing $K_{\mathbf{nope}}$ and $V$ for Improved Joint Low-Rank Compression

In the previous section, we split the key heads into one carrying positional information and the others without positional information, achieving minimal loss. We then apply Principal Component Analysis (PCA) jointly on the values and the non-positional components of the keys (i.e. $K_{nope}$), using activations collected from a small calibration dataset, thereby compressing the projection matrices into a low-rank latent space. However, we observed that although the principal components of the keys were effectively separated with RoRoPE, the norm of the residual key features remained significantly larger than that of the value features. This imbalance caused the direct decomposition to favor principal component directions dominated by the keys.

To mitigate this, we scale $W^{DK}_{\mathrm{nope}}$ by dividing it by

$$\alpha = \frac{\mathbb{E}_t[\|W^{DK}_{\mathrm{nope}}\mathbf{x}_t\|_2]}{\mathbb{E}_t[\|W^{DV}\mathbf{x}_t\|_2]} \tag{11}$$

and correspondingly scale $W^{UK}$ by multiplying it by $\alpha$. Here, $W^{DK}_{\mathrm{nope}} \in \mathbb{R}^{(g-1)d \times D}$ represents the last $g-1$ heads that does not use RoPE in $W^{DK}$.

This transformation is mathematically equivalent and does not affect the overall model outputs, while significantly enhancing the effectiveness of KV cache compression in subsequent steps. More details is provided in the Appendix E and F.

## 5 Experiment

### 5.1 Main Experiment

In this section, we present our main experimental results. Following the experimental setup of MHA2MLA, we converted two models—smolLM 1.7B and Llama 2 7B—into the MLA architecture. We evaluated the models' performance on six benchmarks at three stages: before conversion, immediately after conversion without further training, and after conversion followed by training. For the training process, we used a subset of the pretraining corpus used for the smolLM model. The fine-tuned results of MHA2MLA are taken directly from the original paper. Our experiments were conducted on an 8-GPU machine, each GPU having 40GB of memory and delivering 312 TFLOPS of FP16 compute power. Detailed hyperparameter settings are provided in the Appendix G.

From Table 1, we observe that TransMLA efficiently facilitates architecture migration across various models and KV cache compression ratios. Notably, the untrained performance of MLA models initialized with TransMLA shows minimal degradation in capability compared to the original models—significantly less than the degradation observed with MHA2MLA under equivalent KV cache compression. In fact, using TransMLA to compress the KV cache of Llama 2 7B to just 7.03% of its original size still results in better performance than MHA2MLA's compression to 31.25% on the same model. This highlights the effectiveness of our proposed techniques includes RoRoPE, FreqFold and activation-based balanced KV low-rank factorization.

The low-loss transformation achieved by TransMLA enables us to recover the original model performance with minimal training overhead. As shown in the table, TransMLA achieves stronger performance than MHA2MLA-6B while using significantly fewer training tokens. For instance, when transforming smolLM 1.7B and compressing the KV cache to 31.25% of its original size, we only need 4.9% of the training data used by MHA2MLA and 2 hours training to surpass its performance.

### 5.2 Key Norm Analysis Reveals the Impact of RoRoPE and FreqFold

In this section, we conduct a detailed analysis of the distribution of key activations in the attention module to demonstrate the effectiveness of our proposed methods.

Figure 3a presents the average L2 norm of each key dimension in the first attention layer of the LLaMA 3 8B model, computed on a subset of the WikiText-2 [39] dataset. We compare the original model (in blue), the model transformed using our RoRoPE equivalence method (in orange), and the further approximated model using 4D FreqFold (in green). The top and bottom halves of the plot correspond to pairs of dimensions that share the same rotation angle in RoPE, which we refer to as the real (Re-dim) and imaginary (Im-dim) dimensions.

Table 1: Commonsense reasoning ability of two LLMs with TransMLA compared to MHA2MLA. The six benchmarks include MMLU ([33]), ARC easy and challenge (ARC, [34]), PIQA ([35]), HellaSwag (HS, [36]), OpenBookQA (OBQA, [37]), and Winogrande (WG, [38]). **Tokens** refers to the number of tokens used for further training after the TransMLA conversion. A value of 0 indicates that the model was evaluated immediately after conversion, without any fine-tuning.

| Model | Tokens | KV Mem. | Avg. | MMLU | ARC | PIQA | HS | OBQA | WG |
|---|---|---|---|---|---|---|---|---|---|
| SmolLM-1.7B | 1T | – | 55.90 | 39.27 | 59.87 | 75.73 | 62.93 | 42.80 | 54.85 |
| *- MHA2MLA* | 0 | -68.75% | 40.97 | 27.73 | 41.96 | 63.00 | 29.19 | 34.40 | 49.53 |
| | | -81.25% | 37.14 | 26.57 | 32.73 | 55.77 | 26.90 | 31.40 | 49.49 |
| | | -87.50% | 34.01 | 25.32 | 27.15 | 51.36 | 25.47 | 26.20 | 48.54 |
| | 6B | -68.75% | 54.76 | 38.11 | 57.13 | 76.12 | 61.35 | 42.00 | 53.83 |
| | | -81.25% | 54.65 | 37.87 | 56.81 | 75.84 | 60.41 | 42.60 | 54.38 |
| | | -87.50% | 53.61 | 37.17 | 55.50 | 74.86 | 58.55 | 41.20 | 54.38 |
| *- TransMLA* | 0 | -68.75% | 51.95 | 35.70 | 55.68 | 73.94 | 53.04 | 39.80 | 53.51 |
| | | -81.25% | 47.73 | 32.87 | 47.89 | 69.75 | 48.16 | 36.20 | 51.46 |
| | | -87.50% | 44.12 | 29.97 | 41.72 | 66.87 | 41.15 | 34.80 | 50.28 |
| | 300M | -68.75% | 55.24 | 38.60 | 58.95 | 74.97 | 61.52 | 43.00 | 54.38 |
| | 700M | -81.25% | 54.78 | 37.79 | 57.53 | 75.52 | 59.88 | 42.80 | 55.17 |
| | 1B | -87.50% | 54.01 | 37.24 | 56.32 | 74.81 | 60.08 | 42.40 | 53.20 |
| LLaMA-2-7B | 2T | – | 59.85 | 41.43 | 59.24 | 78.40 | 73.29 | 41.80 | 64.96 |
| *- MHA2MLA* | 0 | -68.75% | 37.90 | 25.74 | 32.87 | 59.41 | 28.68 | 28.60 | 52.09 |
| | | -81.25% | 34.02 | 25.50 | 26.44 | 53.43 | 27.19 | 22.60 | 49.01 |
| | | -87.50% | 32.70 | 25.41 | 25.79 | 50.60 | 26.52 | 19.40 | 48.46 |
| | 6B | -68.75% | 59.51 | 41.36 | 59.51 | 77.37 | 71.72 | 44.20 | 62.90 |
| | | -81.25% | 59.61 | 40.86 | 59.74 | 77.75 | 70.75 | 45.60 | 62.98 |
| | | -87.50% | 58.96 | 40.39 | 59.29 | 77.75 | 69.70 | 43.40 | 63.22 |
| *- TransMLA* | 0 | -68.75% | 58.20 | 39.90 | 57.66 | 77.48 | 70.22 | 41.00 | 62.90 |
| | | -87.50% | 51.19 | 34.39 | 45.38 | 71.27 | 60.73 | 37.40 | 57.93 |
| | | -92.97% | 43.26 | 28.93 | 36.32 | 63.38 | 45.87 | 31.60 | 53.43 |
| | 500M | -68.75% | 59.82 | 40.87 | 59.18 | 77.91 | 71.82 | 45.20 | 63.93 |
| | 3B | -87.50% | 59.36 | 40.77 | 58.84 | 78.18 | 71.28 | 43.60 | 63.46 |
| | 6B | -92.97% | 59.19 | 40.41 | 58.68 | 77.53 | 70.39 | 45.00 | 63.14 |
| LLaMA-3-8B | 15T | – | 63.84 | 46.20 | 65.75 | 80.47 | 76.20 | 45.60 | 68.82 |
| *- TransMLA* | 0 | -71.875% | 54.13 | 36.38 | 52.84 | 73.83 | 64.34 | 37.00 | 60.38 |
| | 30B | -71.875% | 63.39 | 46.18 | 66.30 | 80.30 | 76.33 | 45.00 | 66.22 |
| | 60B | -71.875% | 63.76 | 47.39 | 66.96 | 80.41 | 77.10 | 44.80 | 65.90 |

We observe that the original model exhibits a highly *uneven norm distribution* across key dimensions, with numerous outliers. This suggests that naively removing RoPE from certain heads would likely result in significant performance degradation. After applying our RoRoPE transformation, as shown by the orange line, the key dimensions with large norms are nearly all concentrated to the first two heads (dimension 0-128). Further applying the 4D FreqFold approximation compresses the tail (i.e., higher-index dimensions) even more, leading to an even sharper concentration of high-norm key dimensions. This concentrated structure is highly beneficial for the subsequent RoPE removal step as shown in Figure 3b.

In Figure 3b, we present the log-perplexity of the LLaMA 3 8B model on WikiText-2 as RoPE components are progressively removed. We observe that our proposed RoRoPE method *significantly outperforms* MHA2MLA's per-head dimension selection strategy, especially at high removal ratio. Furthermore, incorporating similar-dimension approximation leads to even better performance under extreme removal rates. At 90% removal ration, RoRoPE + 4D-FreqFold still maintains a log-perplexity about 2, while MHA2MLA reaches nearly 6, which no longer generates meaningful outputs. At the same time, we observe that overly aggressive FreqFold (i.e., using too many dimensions) can degrade performance, as the loss introduced by approximation of nearby dimensions can outweigh

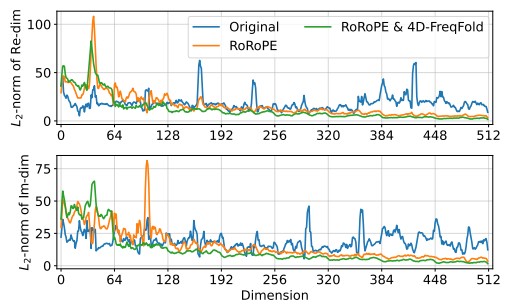

(a) Key norms of the first layer.

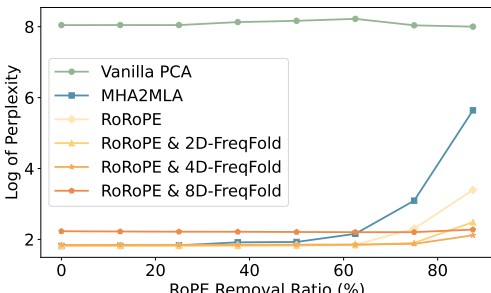

(b) RoPE removal results, evaluated on WikiText-2.

Figure 3: Visualization of key norms and RoPE removal results on LLaMA 3 8B model. The top and bottom halves of the left figure correspond to pairs of dimensions that share the same rotation angle in RoPE, which we refer to as the real (Re-dim) and imaginary (Im-dim) dimensions.

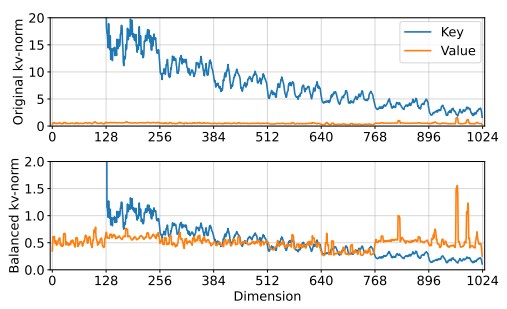

(a) Norm magnitude of keys and values before and after KV balancing is applied.

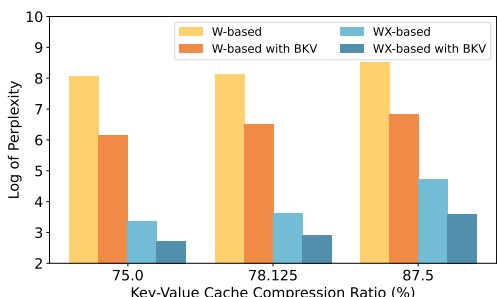

(b) Comparison of different methods of KV cache compression, with and without KV balancing.

Figure 4: Visualization of the norms of keys and values for the first layer of LLaMA 3 8B and the perplexity results after joint low-rank compression of keys and values under WikiText-2. Here, **W-based** and **WX-based** refer to PCA applied on the attention weights and the activation outputs, respectively. **BKV** denotes the application of KV balancing.

the benefit in concentrating the principal components. This figure suggests that for LLaMA 3 8B, the sweet spot lies in applying RoRoPE combined with 4D FreqFold.

## 5.3 Key-Value Norm Disparity Motivates KV Balancing

In Figure 4a, we visualize the norm magnitudes of the key and value activations in the first layer of LLaMA 3 8B before and after KV balancing. Note that both the key and value shown here are activations after applying the RoRoPE principal component concentration, and the first head of the key—reserved for RoPE—is excluded. As a result, the value and the remaining key components shown in the figure are precisely the elements we aim to compress jointly into a lower-dimensional space via PCA.

It is evident that even after removing the first head with the highest norm, the overall norm of the key remains significantly larger than that of the value. This norm disparity poses a substantial challenge for joint compression into a shared latent space. In particular, such imbalance can bias the PCA toward directions aligned with the key, rather than the value, leading to suboptimal representation of the value components.

The lower part of Figure 4a shows the norm distribution of keys and values after applying KV balancing. At this point, the norms of keys and values become more aligned, which is beneficial for performing joint PCA. This observation is further supported by the results in Figure 4b, where KV balancing consistently reduces the loss incurred by jointly applying low-rank approximation to keys and values—whether the PCA is based on weights or on activations. Figure 4b also demonstrates that activation-based PCA yields significantly better results than weight-based PCA.

## 5.4  Hardware-Agnostic Inference Speedup with TransMLA

By converting MHA/GQA models into MLA models that are fully compatible with the DeepSeek codebase and compressing the KV cache, TransMLA enables us to leverage all optimizations and tooling available in DeepSeek. Using the vLLM framework, we achieve substantial real-world inference speedups.

In Figure 5, we benchmarked the inference performance of an MLA model—with a 92.97% reduction in KV cache size—on three consumer-grade AI accelerators with different compute capabilities and memory sizes: 165.2 TFLOPS with 24GB memory, 312 TFLOPS with 40GB memory, and 320 TFLOPS with 64GB memory. The figure shows the inference speedup of the MLA model relative to the original MHA model. Low-rank Q and Full-rank Q indicate whether the query projections were also compressed. Context length represents the total sequence length (i.e., context length plus generated tokens).

Our experiments show that the inference speedup of MLA models increases as the context length grows, which aligns with our expectations. Since the primary performance gain of MLA stems from KV cache compression, longer contexts lead to more substantial savings and thus higher speedups. Remarkably, for the 8K context window on the first hardware platform, the TransMLA-transformed model achieves an impressive **10x inference acceleration**. To the best of our knowledge, the MHA2MLA method has not reported any inference speedup results.

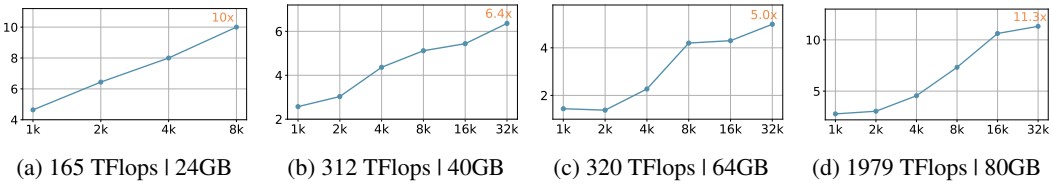

(a) 165 TFlops | 24GB    (b) 312 TFlops | 40GB    (c) 320 TFlops | 64GB    (d) 1979 TFlops | 80GB

Figure 5: Inference speedups with TransMLA comparing to the original LLaMA2 7B model on three consumer-grade AI accelerators. **Context length** represents the total sequence length.

## 6  Conclusion, Limitation and Future Work

In this work, we demonstrate that the expressive power of TransMLA is stronger than GQA under the same KV cache. To help existing GQA transition to the MLA structure with minimal cost, we propose the TransMLA method. By using the RoRoPE method, the multi-head KV positional information is concentrated into the first head, and FreqFold further enhances this extraction effect. The positional information of the remaining query-key heads is removed, and the Balance KV norm method is used to jointly compress the values and the remaining heads of keys. TransMLA can convert models such as LLaMA and Qwen into MLA-based models, and the converted model incurs very little loss compared to the original model, with performance being recoverable through training with only a few tokens. Additionally, TransMLA can easily leverage the DeepSeek ecosystem for accelerated inference, achieving significant throughput improvements across various hardware platforms.

Although TransMLA significantly reduces the loss caused by decoupling RoPE through RoRoPE and FreqFold, and mitigates KV cache compression loss via KV balancing, error accumulation becomes increasingly severe for longer texts, making recovery more challenging. Moreover, potential issues may arise during training aimed at enhancing recovery performance, including subtle performance degradation and hallucinations, which must be carefully considered. To address these challenges, it is worth exploring more advanced mathematical approaches that can better reconcile the norm disparity between keys and values, achieving improved performance under the same compression rate and ultimately enabling a truly training-free conversion process.

All speed comparisons in this paper are conducted on the single GPU. When multi-GPU parallel inference is required, MLA is less compatible with tensor parallelism (TP) than GQA, since multiple devices must replicate the same latent cache, which typically necessitates the use of data parallelism (DP). How can the speed of TransMLA be improved under TP? Furthermore, TransMLA should be integrated with pruning, quantization, token selection, and other optimization techniques to fully explore the upper limits of inference acceleration.

# 7   Acknowledgements

This work is supported by the National Key R&D Program of China (2022ZD0160300), Center of Excellence, Peking University, and CCF-Tencent Rhino-Bird Open Research Fund.

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

# Contents

# A  Additional Preliminaries

## A.1  Rotary Position Embedding (RoPE)

RoPE [22] is a position encoding method that encodes the absolute positions with different rotations and incorporates the explicit relative position dependency in the self-attention formulation. It applies different rotations to tokens in different positions to encode the position information.

Consider $\mathbf{x}_t \in \mathbb{R}^d$ to be the embedding of the $t$-th token with the hidden size $d$. The RoPE operation upon $\mathbf{x}_t$ produces a representation $\mathbf{x}_t^R$ that encodes both semantic and positional information:

$$
\mathbf{x}_t^R = \mathrm{RoPE}_t(\mathbf{x}_t) = 
\begin{pmatrix} \mathbf{x}_t^{(1)} \\ \mathbf{x}_t^{(2)} \\ \mathbf{x}_t^{(3)} \\ \mathbf{x}_t^{(4)} \\ \vdots \\ \mathbf{x}_t^{(d-1)} \\ \mathbf{x}_t^{(d)} \end{pmatrix}
\otimes
\begin{pmatrix} \cos t\theta_1 \\ \cos t\theta_1 \\ \cos t\theta_2 \\ \cos t\theta_2 \\ \vdots \\ \cos t\theta_{d/2} \\ \cos t\theta_{d/2} \end{pmatrix}
+
\begin{pmatrix} -\mathbf{x}_t^{(2)} \\ \mathbf{x}_t^{(1)} \\ -\mathbf{x}_t^{(4)} \\ \mathbf{x}_t^{(3)} \\ \vdots \\ -\mathbf{x}_t^{(d)} \\ \mathbf{x}_t^{(d-1)} \end{pmatrix}
\otimes
\begin{pmatrix} \sin t\theta_1 \\ \sin t\theta_1 \\ \sin t\theta_2 \\ \sin t\theta_2 \\ \vdots \\ \sin t\theta_{d/2} \\ \sin t\theta_{d/2} \end{pmatrix},
\tag{12}
$$

where $\otimes$ denotes the element-wise multiplication of two vectors, $\mathbf{x}_t^{(i)} \in \mathbb{R}$ denotes the $i$-th element of $\mathbf{x}_t$, and $\theta_i = 10000^{-2(i-1)/d}$ is the $i$-th rotation angle. If we interpret every two elements in the embedding as a representation in the complex coordinate system, we can divide $\mathbf{x}_t$ into paired dimensions, where the odd-indexed dimensions $\mathbf{x}_t^{(2k-1)}$ represent the **real parts** and the even-indexed dimensions $\mathbf{x}_t^{(2k)}$ represent the **imaginary parts**.

## A.2  Group Query Attention

Let the $t$-th token of the input sequence be $\mathbf{x}_t \in \mathbb{R}^D$, where $D$ denotes the hidden dimension. To reduce the memory overhead of the KV cache, GQA divides the $h$ query heads uniformly into $g$ groups, with all query heads within a group sharing the same key and value vectors. Specifically, let $W^Q \in \mathbb{R}^{hd \times D}$, $W^K, W^V \in \mathbb{R}^{gd \times D}$ and $W^O \in \mathbb{R}^{D \times hd}$ be the projection matrices for the query, key, value and output, where $d = D/h$ denotes the dimension per head. GQA first computes the concatenated queries $\mathbf{q}_t$, keys $\mathbf{k}_t$, and values $\mathbf{v}_t$, and then slices them into heads or groups for attention computation:

$$[\mathbf{q}_{t,1}; \mathbf{q}_{t,2}; ...; \mathbf{q}_{t,h}] = \mathbf{q}_t = W^Q \mathbf{x}_t, \tag{13}$$

$$[\mathbf{k}_{t,1}; \mathbf{k}_{t,2}; ...; \mathbf{k}_{t,g}] = \mathbf{k}_t = W^K \mathbf{x}_t, \tag{14}$$

$$[\mathbf{v}_{t,1}; \mathbf{v}_{t,2}; ...; \mathbf{v}_{t,g}] = \mathbf{v}_t = W^V \mathbf{x}_t, \tag{15}$$

where each $\mathbf{q}_{t,i} \in \mathbb{R}^d$ corresponds to the query vector of the i-th head, and $\mathbf{k}_{t,j}, \mathbf{v}_{t,j} \in \mathbb{R}^d$ correspond to the key and value vectors of the j-th group.

Using the notation in Section A.1, after applying RoPE to $\mathbf{q}_{t,i}, \mathbf{k}_{t,i}$, we can obtain the attention output for the $t$-th token as follows:

$$\mathbf{o}_{t,i} = \sum_{j=1}^{t} \mathrm{softmax}_j \left( \frac{\mathbf{q}_{t,i}^{R\ \top} \mathbf{k}_{j,\lceil i/\frac{h}{g}\rceil}^R}{\sqrt{d}} \right) \mathbf{v}_{j,\lceil i/\frac{h}{g}\rceil}, \tag{16}$$

$$\mathbf{y}_t = W^O [\mathbf{o}_{t,1}; \mathbf{o}_{t,2}; ...; \mathbf{o}_{t,h}]. \tag{17}$$

As we can see, in GQA, each key and value head corresponds to $\frac{h}{g}$ query heads. **When $g = h$, GQA becomes MHA, and when $g = 1$, GQA becomes Multi-Query Attention (MQA).**

# B  Enhanced Expressive Power of MLA with Decoupled RoPE

## B.1  Introduction

This section provides a theoretical analysis to demonstrate that Multi-Head Latent Attention (MLA) with decoupled Rotary Position Embedding (RoPE), as described in Section 3 of the main paper, possesses greater expressive power than Grouped-Query Attention (GQA) (Section A.2). This analysis assumes **comparable KV cache sizes and number of query heads**.

Our primary argument focuses on the core projection mechanisms that generate queries, keys, and values, abstracting away from the specifics of RoPE application initially. We first present the following proposition concerning the relative expressiveness of these core mechanisms:

**Proposition 1.** *Given the same KV cache size and number of query heads, the expressiveness of the core attention projection mechanisms follows the order:* $\mathrm{GQA} < \mathrm{MLA}_{\mathrm{Factorized}} < \mathrm{MQA}$.

Here, $\mathrm{MLA}_{\mathrm{Factorized}}$ refers to an attention mechanism employing low-rank factorization for its key and value projections, representing the content-processing aspect of the full MLA. It is important to note that in the proposition, the query projection in $\mathrm{MLA}_{\mathrm{Factorized}}$ does not undergo low-rank factorization; this differs from the full MLA, where the query is also factorized. After proving this proposition, we will discuss how the full MLA architecture, which incorporates such an $\mathrm{MLA}_{\mathrm{Factorized}}$ core for its content components and an $\mathrm{MQA}$ core for its decoupled RoPE components, is thereby more expressive than GQA. For this analysis, we primarily consider the impact of the architectural structure on representational capacity, **setting aside the direct effects of RoPE itself** on the expressiveness comparison between the fundamental GQA, MLA-Factorized, and MQA structures.

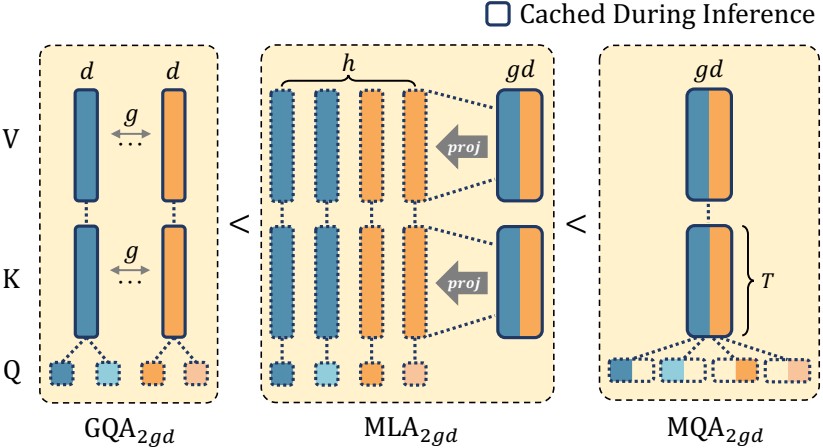

Figure 6: Comparison of Multi-Query Attention (MQA), Group Query Attention (GQA), and Multi-Head Latent Attention (MLA). In this work, we illustrate that given the same KV cache size, the expressiveness increases in the order of GQA, MLA, and MQA. In the figure, $h, d, g$ denote the number of heads, hidden dimension of each head, and the number of groups (K/V heads) in GQA, respectively. In MQA, the head dimension is set to $gd$ to align the KV cache size with GQA and MLA. As a result, the KV cache size per token per layer for all three approaches is $2gd$

## B.2  Proof of Proposition 1

Let $D$ be the hidden dimension of the input token $\mathbf{x}_t \in \mathbb{R}^D$, $h$ be the number of query heads, and $d = D/h$ be the dimension per head. In GQA, query heads are divided into $g$ groups. For fair KV cache comparison, the latent dimension for keys and values in $\mathrm{MLA}_{\mathrm{Factorized}}$ ($r_{kv}$) and the head dimension of MQA will be related to $gd$. Specifically, if the KV cache per token in GQA is $2gd$ for both keys and values, then in $\mathrm{MLA}_{\mathrm{Factorized}}$, $r_{kv} = 2gd$, and in MQA, the head dimension is also $2gd$; this ensures the KV cache sizes are aligned.

### B.2.1 GQA $\leq$ MLA$_{\text{Factorized}}$

In GQA, query head $\mathbf{q}_{t,i}$ attends to key $\mathbf{k}_{j,\lceil i/(h/g)\rceil}$ and value $\mathbf{v}_{j,\lceil i/(h/g)\rceil}$. The GQA key projection $W^K \in \mathbb{R}^{gd \times D}$ produces $g$ distinct key vectors $[\mathbf{k}_{t,1}; \ldots; \mathbf{k}_{t,g}]$. Similarly, $W^V \in \mathbb{R}^{gd \times D}$ produces value vectors. We define effective per-query-head projection matrices $W'^K \in \mathbb{R}^{hd \times D}$ and $W'^V \in \mathbb{R}^{hd \times D}$ for GQA:

$$W'^K = \begin{pmatrix} W'^K_1 \\ \vdots \\ W'^K_h \end{pmatrix}, \text{where } W'^K_i = W^K_{\lceil i/(h/g)\rceil}, \tag{18}$$

$$W'^V = \begin{pmatrix} W'^V_1 \\ \vdots \\ W'^V_h \end{pmatrix}, \text{where } W'^V_i = W^V_{\lceil i/(h/g)\rceil}. \tag{19}$$

Here, $W^K_k$ is the $k$-th $d \times D$ block of $W^K$. Thus, $\mathbf{k}'_{j,i} = W'^K_i \mathbf{x}_j = \mathbf{k}_{j,\lceil i/(h/g)\rceil}$, and similarly for values. The matrices $W'^K$ and $W'^V$ have ranks at most $gd$.

An MLA$_{\text{Factorized}}$ mechanism generates keys via $\mathbf{k}_{j,i} = (W^{UK}(W^{DKV}\mathbf{x}_j))_i$, where $W^{DKV} \in \mathbb{R}^{r_{kv} \times D}$ and $W^{UK} \in \mathbb{R}^{hd \times r_{kv}}$. A similar formulation applies for values with $W^{UV} \in \mathbb{R}^{hd \times r_{kv}}$.

To demonstrate expressive capability, GQA $\leq$ MLA$_{\text{Factorized}}$, we set $r_{kv} = 2gd$. Let $W^{DKV} = \begin{pmatrix} W^K \\ W^V \end{pmatrix} \in \mathbb{R}^{2gd \times D}$. We seek $W^{UK}, W^{UV} \in \mathbb{R}^{hd \times 2gd}$ such that $W'^K = W^{UK}W^{DKV}, W'^V = W^{UV}W^{DKV}$. This is achieved by setting $W^{UK}_i, W^{UV}_i \in \mathbb{R}^{d \times 2gd}$ (the block for head $i$) as selector matrices:

$$W^{UK}_i = [\underbrace{\mathbf{0}_{d \times d}, \ldots, \mathbf{0}_{d \times d}}_{k-1 \text{ blocks}}, \mathbf{I}_{d \times d}, \underbrace{\mathbf{0}_{d \times d}, \ldots, \mathbf{0}_{d \times d}}_{2g-k \text{ blocks}}], \tag{20}$$

$$W^{UV}_i = [\underbrace{\mathbf{0}_{d \times d}, \ldots, \mathbf{0}_{d \times d}}_{g+k-1 \text{ blocks}}, \mathbf{I}_{d \times d}, \underbrace{\mathbf{0}_{d \times d}, \ldots, \mathbf{0}_{d \times d}}_{g-k \text{ blocks}}], \tag{21}$$

where $k = \lceil i/(h/g)\rceil$. Thus, GQA's key/value generation can be replicated by an MLA$_{\text{Factorized}}$ model with $r_{kv} = 2gd$ and specific sparse structures for $W^{UK}$ and $W^{UV}$. The KV cache size $2gd \times$ (sequence length) is preserved since we will be caching $\mathbf{c}^{KV}_j = W^{DKV}\mathbf{x}_j \in \mathbb{R}^{2gd}$. On that account, the theoretical expressive power of GQA is less than or equal to that of MLA$_{\text{Factorized}}$ given the same KV cache size.

### B.2.2 MLA$_{\text{Factorized}}$ $\leq$ MQA

Consider an MLA-Factorized model where queries are $\mathbf{q}_{t,i} = W^Q_i \mathbf{x}_t$ (assuming $W^Q_i \in \mathbb{R}^{d \times D}$ is the $i$-th block of $W^Q$) and keys are $\mathbf{k}_{j,i} = (W^{UK}_i(W^{DKV}\mathbf{x}_j))$. The attention score for head $i$ involves $\mathbf{q}^\top_{t,i}\mathbf{k}_{j,i}$:

$$\mathbf{q}^\top_{t,i}\mathbf{k}_{j,i} = (W^Q_i \mathbf{x}_t)^\top (W^{UK}_i(W^{DKV}\mathbf{x}_j)). \tag{22}$$

This can be rewritten as:

$$\mathbf{q}^\top_{t,i}\mathbf{k}_{j,i} = (\underbrace{(W^{UK}_i)^\top W^Q_i}_{W'^Q_i} \mathbf{x}_t)^\top (W^{DKV}\mathbf{x}_j). \tag{23}$$

Let $\hat{\mathbf{q}}_{t,i} = W_i'^Q \mathbf{x}_t \in \mathbb{R}^{2gd}$ and $\mathbf{c}_j^{KV} = W^{DKV} \mathbf{x}_j \in \mathbb{R}^{2gd}$. The computation of attention output becomes:

$$\mathbf{o}_{t,i} = \sum_j \text{softmax}_j\left(\frac{\hat{\mathbf{q}}_{t,i}^\top \mathbf{c}_j^{KV}}{\sqrt{d}}\right) W_i^{UV} \mathbf{c}_j^{KV}, \tag{24}$$

$$\mathbf{y}_t = W^O [\mathbf{o}_{t,1}; \mathbf{o}_{t,2}; ...; \mathbf{o}_{t,h}]$$

$$= W^O \underbrace{\begin{pmatrix} W_1^{UV} & & & \\ & W_2^{UV} & & \\ & & \ddots & \\ & & & W_h^{UV} \end{pmatrix}}_{W'^O} \begin{pmatrix} \text{softmax}_j\left(\frac{\hat{\mathbf{q}}_{t,1}^\top \mathbf{c}_j^{KV}}{\sqrt{d}}\right)\mathbf{c}_j^{KV} \\ \vdots \\ \text{softmax}_j\left(\frac{\hat{\mathbf{q}}_{t,1}^\top \mathbf{c}_j^{KV}}{\sqrt{d}}\right)\mathbf{c}_j^{KV} \end{pmatrix}. \tag{25}$$

This is an MQA formulation where each modified query $\hat{\mathbf{q}}_{t,i}$ (now of dimension $2gd$) attends to a shared key and value $\mathbf{c}_j^{KV}$. This indicates that the computations within MLA-Factorized can be structured to use shared intermediate key and value representations akin to MQA's core. Thus, any MLA-Factorized model can be represented as an MQA model with a shared key/value of dimension $2gd$.

### B.2.3 Strict Inequalities: GQA < MLA$_\text{Factorized}$ < MQA

The relationships are strict:

**GQA < MLA$_\text{Factorized}$**   When GQA is represented as an MLA$_\text{Factorized}$ model, the up-projection matrices $W^{UK}$ and $W^{UV}$ must adopt specific sparse, block-selector structures. A general MLA$_\text{Factorized}$ model imposes no such constraints; $W^{UK}$ and $W^{UV}$ are typically dense and fully learnable. This allows a general MLA$_\text{Factorized}$ to create $h$ distinct key (and value) vectors by combining features from the $r_{kv}$-dimensional latent space in complex ways. GQA is restricted to $g$ unique key (and value) vectors that are merely replicated $h/g$ times. If $h > g$, MLA$_\text{Factorized}$ can generate a richer set of interaction patterns. Thus, MLA$_\text{Factorized}$ has strictly greater expressive power.

**MLA$_\text{Factorized}$ < MQA**   Consider the bilinear form $\mathbf{x}_t^\top \mathbf{M} \mathbf{x}_j$ in the attention score. In MLA$_\text{Factorized}$, for head $i$, $\mathbf{M}_{MLA,i} = (W_i^Q)^\top W_i^{UK} W^{DKV}$. The maximum rank of the transformation is determined by the smallest one among the ranks of $W_i^Q \in \mathbb{R}^{d \times D}$, $W_i^{UK} \in \mathbb{R}^{d \times 2gd}$, and $W^{DKV} \in \mathbb{R}^{2gd \times D}$, which is at most $d$.

However, in the MQA form derived from MLA$_\text{Factorized}$, the rank of the interaction matrix here, $(W_i'^Q)^\top W^{DKV}$, is determined by the smallest one among the ranks of $W_i'^Q \in \mathbb{R}^{2gd \times D}$ and $W^{DKV} \in \mathbb{R}^{2gd \times D}$, which is at most $2gd$.

Since $2gd \geq d$, MQA allows for a potentially higher-rank interaction between the (modified) query and the shared key representations compared to the per-head effective rank in MLA$_\text{Factorized}$'s original formulation. This indicates that MQA has a greater representational capacity for the scoring mechanism.

### B.3 Expressiveness of MLA with Decoupled RoPE

The full MLA architecture, as defined in Section 3 (main paper), employs a decoupled RoPE strategy. The query $\mathbf{q}_{t,i}$ and key $\mathbf{k}_{t,i}$ for head $i$ (in the MHA-like training paradigm, Equation 3) are:

$$\mathbf{q}_{t,i} = [\mathbf{q}_{t,i}^C; \mathbf{q}_{t,i}^R] \tag{26}$$

$$\mathbf{k}_{t,i} = [\mathbf{k}_{t,i}^C; \mathbf{k}_t^R] \tag{27}$$

where $\mathbf{k}_t^R$ is a shared RoPE key component across all heads for token $t$. The bilinear attention score (numerator of the softmax argument) for head $i$ between query at $t$ and key at $j$ is:

$$(\mathbf{q}_{t,i}^C)^\top \mathbf{k}_{j,i}^C + (\mathbf{q}_{t,i}^R)^\top \mathbf{k}_j^R \tag{28}$$

Let's analyze the two components of this score:

1. **Content Component Interaction**: $(\mathbf{q}_{t,i}^C)^\top \mathbf{k}_{j,i}^C$. The content keys $\mathbf{k}_{j,i}^C$ are derived from $W^{UK}(W^{DKV}\mathbf{x}_j)$. This key generation mechanism for $\mathbf{k}_{j,i}^C$ is precisely that of the $\mathrm{MLA_{Factorized}}$ model discussed in Section B.1. As established, MLA-Factorized is strictly more expressive than GQA for the non-positional part of the representation.

2. **Positional Component Interaction**: $(\mathbf{q}_{t,i}^R)^\top \mathbf{k}_j^R$. This interaction, where $h$ distinct query-side RoPE components $\mathbf{q}_{t,i}^R$ attend to a single, shared key-side RoPE component $\mathbf{k}_j^R$, is an MQA structure specifically for the positional information. As shown in Section B.2.3, MQA is strictly more expressive than $\mathrm{MLA_{Factorized}}$, and by extension, GQA.

In summary, we have demonstrated that the expressive power of MLA with decoupled RoPE is stronger than that of the traditional GQA. However, it is worth noting that in the previously proven proposition, the $\mathrm{MLA_{Factorized}}$ does not have a low-rank decomposition on the query; this differs from DeepSeek MLA. In the full MLA architecture, the query is also decomposed.

# C  Proof of RoPE Inner Product Invariance under Orthogonal Transformation

In this subsection, we provide a rigorous proof of Equation (10), namely:

$$\sum_{l=1}^{d/2} \overline{\mathrm{RoPE}}_{t,l}\left(\left[\mathbf{U}_l\hat{\mathbf{q}}_{t,i}^{(l,\mathrm{real})}; \mathbf{U}_l\hat{\mathbf{q}}_{t,i}^{(l,\mathrm{imag})}\right]\right)^\top \overline{\mathrm{RoPE}}_{j,l}\left(\left[\mathbf{U}_l\hat{\mathbf{k}}_j^{(l,\mathrm{real})}; \mathbf{U}_l\hat{\mathbf{k}}_j^{(l,\mathrm{imag})}\right]\right) = \hat{\mathbf{q}}_{t,i}^{R\top}\hat{\mathbf{k}}_j^R.$$

Here, $d$ is the dimension of each original attention head. The notation $\hat{\mathbf{q}}_{t,i}^{(l,\mathrm{real})}$ (and similarly for other terms) refers to an $g$-dimensional vector collecting the $(2l-1)$-th components from each of the $g$ original attention heads. The matrix $\mathbf{U}_l$ is an $g \times g$ orthogonal matrix.

*Proof.* For the sake of convenience, we omit all $i, j, k$ and let $\mathbf{q}_{x,l} = \hat{\mathbf{q}}_{t,i}^{(l,\mathrm{real})}$ and $\mathbf{q}_{y,l} = \mathbf{q}_{t,i}^{(l,\mathrm{imag})}$. Similarly, let $\mathbf{k}_{x,l} = \hat{\mathbf{k}}_j^{(l,\mathrm{real})}$ and $\mathbf{k}_{y,l} = \hat{\mathbf{k}}_j^{(l,\mathrm{imag})}$.

The RoPE transformation, as defined by Equation (9) in the main text, applies as follows for a query vector at position $t$ and key vector at position $j$ within the $l$-th subspace:

$$(\mathbf{q}_{x,l})^R = \mathbf{q}_{x,l}\cos(t\theta_l) - \mathbf{q}_{y,l}\sin(t\theta_l)$$
$$(\mathbf{q}_{y,l})^R = \mathbf{q}_{x,l}\sin(t\theta_l) + \mathbf{q}_{y,l}\cos(t\theta_l)$$
$$(\mathbf{k}_{x,l})^R = \mathbf{k}_{x,l}\cos(j\theta_l) - \mathbf{k}_{y,l}\sin(j\theta_l)$$
$$(\mathbf{k}_{y,l})^R = \mathbf{k}_{x,l}\sin(j\theta_l) + \mathbf{k}_{y,l}\cos(j\theta_l)$$

The superscript $R$ denotes the application of RoPE. We use the shorthand $c_t = \cos(t\theta_l)$, $s_t = \sin(t\theta_l)$, $c_j = \cos(j\theta_l)$, and $s_j = \sin(j\theta_l)$.

The right-hand side (RHS) of Equation (10) is given by the definition of the RoPE inner product:

$$\mathbf{q}_{t,i}^{R\top}\mathbf{k}_j^R = \sum_{l=1}^{d/2}\left[(\mathbf{q}_{x,l})^R; (\mathbf{q}_{y,l})^R\right]^\top\left[(\mathbf{k}_{x,l})^R; (\mathbf{k}_{y,l})^R\right]$$
$$= \sum_{l=1}^{d/2}((\mathbf{q}_{x,l})^R)^\top(\mathbf{k}_{x,l})^R + ((\mathbf{q}_{y,l})^R)^\top(\mathbf{k}_{y,l})^R$$

Let $S_l$ be the $l$-th term in this sum:

$$S_l = (c_t\mathbf{q}_{x,l} - s_t\mathbf{q}_{y,l})^\top(c_j\mathbf{k}_{x,l} - s_j\mathbf{k}_{y,l}) + (s_t\mathbf{q}_{x,l} + c_t\mathbf{q}_{y,l})^\top(s_j\mathbf{k}_{x,l} + c_j\mathbf{k}_{y,l})$$
$$= c_tc_j\mathbf{q}_{x,l}^\top\mathbf{k}_{x,l} - c_ts_j\mathbf{q}_{x,l}^\top\mathbf{k}_{y,l} - s_tc_j\mathbf{q}_{y,l}^\top\mathbf{k}_{x,l} + s_ts_j\mathbf{q}_{y,l}^\top\mathbf{k}_{y,l}$$
$$\quad + s_ts_j\mathbf{q}_{x,l}^\top\mathbf{k}_{x,l} + s_tc_j\mathbf{q}_{x,l}^\top\mathbf{k}_{y,l} + c_ts_j\mathbf{q}_{y,l}^\top\mathbf{k}_{x,l} + c_tc_j\mathbf{q}_{y,l}^\top\mathbf{k}_{y,l}$$
$$= (c_tc_j + s_ts_j)(\mathbf{q}_{x,l}^\top\mathbf{k}_{x,l} + \mathbf{q}_{y,l}^\top\mathbf{k}_{y,l}) + (s_tc_j - c_ts_j)(\mathbf{q}_{x,l}^\top\mathbf{k}_{y,l} - \mathbf{q}_{y,l}^\top\mathbf{k}_{x,l})$$
$$= \cos((t-j)\theta_l)(\mathbf{q}_{x,l}^\top\mathbf{k}_{x,l} + \mathbf{q}_{y,l}^\top\mathbf{k}_{y,l}) + \sin((t-j)\theta_l)(\mathbf{q}_{x,l}^\top\mathbf{k}_{y,l} - \mathbf{q}_{y,l}^\top\mathbf{k}_{x,l}).$$

Now, let's analyze the left-hand side (LHS) of Equation (10). Let $\mathbf{q}'_{x,l} = \mathbf{U}_l\mathbf{q}_{x,l}$ and $\mathbf{q}'_{y,l} = \mathbf{U}_l\mathbf{q}_{y,l}$. Similarly, let $\mathbf{k}'_{x,l} = \mathbf{U}_l\mathbf{k}_{x,l}$ and $\mathbf{k}'_{y,l} = \mathbf{U}_l\mathbf{k}_{y,l}$. The $l$-th term of the LHS sum, denoted $S'_l$, is:

$$S'_l = \left( ((\mathbf{q}'_{x,l})^R)^\top (\mathbf{k}'_{x,l})^R + ((\mathbf{q}'_{y,l})^R)^\top (\mathbf{k}'_{y,l})^R \right).$$

This has the same structure as $S_l$, just with primed variables:

$$S'_l = \cos((t-j)\theta_l)((\mathbf{q}'_{x,l})^\top \mathbf{k}'_{x,l} + (\mathbf{q}'_{y,l})^\top \mathbf{k}'_{y,l}) + \sin((t-j)\theta_l)((\mathbf{q}'_{x,l})^\top \mathbf{k}'_{y,l} - (\mathbf{q}'_{y,l})^\top \mathbf{k}'_{x,l}).$$

We need to show that the dot product terms involving primed variables are equal to their unprimed counterparts. Consider the first coefficient term:

$$
\begin{aligned}
(\mathbf{q}'_{x,l})^\top \mathbf{k}'_{x,l} + (\mathbf{q}'_{y,l})^\top \mathbf{k}'_{y,l} &= (\mathbf{U}_l\mathbf{q}_{x,l})^\top (\mathbf{U}_l\mathbf{k}_{x,l}) + (\mathbf{U}_l\mathbf{q}_{y,l})^\top (\mathbf{U}_l\mathbf{k}_{y,l}) \\
&= \mathbf{q}_{x,l}^\top \mathbf{U}_l^\top \mathbf{U}_l \mathbf{k}_{x,l} + \mathbf{q}_{y,l}^\top \mathbf{U}_l^\top \mathbf{U}_l \mathbf{k}_{y,l} \\
&= \mathbf{q}_{x,l}^\top \mathbf{k}_{x,l} + \mathbf{q}_{y,l}^\top \mathbf{k}_{y,l}.
\end{aligned}
$$

The last equation holds because $U_l$ is an orthogonal matrix. This matches the corresponding term in $S_l$.

The same applies to the second coefficient term. In this way, we have proven that $S'_l = S_l$ for each $l \in \{1, \ldots, d/2\}$. This implies that the LHS of Equation (10) is equal to its RHS:

$$\sum_{l=1}^{d/2} \left( \left[ \mathbf{U}_l\mathbf{q}_{t,i}^{[2l-1::]}; \mathbf{U}_l\mathbf{q}_{t,i}^{[2l::]} \right] \right)^{R\top} \left( \left[ \mathbf{U}_l\mathbf{k}_j^{[2l-1::]}; \mathbf{U}_l\mathbf{k}_j^{[2l::]} \right] \right)^R = \mathbf{q}_{t,i}^{R\top} \mathbf{k}_j^R.$$

This completes the proof, demonstrating that the orthogonal transformation $\mathbf{U}_l$ applied to the $g$-dimensional vectors representing the $l$-th 2D subspace components across heads preserves the RoPE-based inner product structure. $\square$

In practice, we leverage this rotational invariance property to find a set of optimal orthogonal matrices $\{\mathbf{U}_l\}$ that concentrate the principal components of the key vectors into the first few attention heads. The preceding proof reveals a critical constraint: for the inner product's value to remain unchanged after transformation, the same orthogonal matrix $\mathbf{U}_l$ must be applied to both the real ($2l - 1$) and imaginary ($2l$) components of the key vectors within each 2D subspace. This requirement precludes performing separate PCA on the real and imaginary parts. We must therefore find a single rotation that is jointly optimal for both.

Specifically, we formulate this as a joint optimization problem. First, we process a calibration dataset (e.g., Wikitext-2) to collect the key activations at each layer. For each RoPE subspace $l \in \{1, \ldots, d/2\}$, we obtain two collections of $n \times g$-dimensional matrices (where $n$ denotes the number of samples): the "real" parts $\{\mathbf{K}_{x,l}\}_l$ and the "imaginary" parts $\{\mathbf{K}_{y,l}\}_l$. To find a single transformation $U_l$ that simultaneously compresses the information from both sets into the first few heads, we proceed as follows.

Let $\sigma_{x,l} = \mathbf{K}_{x,l}^\top \mathbf{K}_{x,l}$ and $\sigma_{y,l} = \mathbf{K}_{y,l}^\top \mathbf{K}_{y,l}$ be the $g \times g$ covariance matrices of the real and imaginary key components, respectively. Our objective is to find an orthogonal matrix $\mathbf{U}_l$ that maximizes the variance—or energy—concentrated in the first $m$ heads after rotation. This corresponds to maximizing the trace of the top-left $m \times m$ submatrix of the *summed* covariance of the rotated vectors. The problem is formally stated as:

$$\max_{\mathbf{U}_l} \quad \mathrm{Tr}\left[ (\mathbf{U}_l^T(\sigma_{x,l} + \sigma_{y,l})\mathbf{U}_l)_{:m,:m} \right] \quad \text{s.t.} \quad \mathbf{U}_l^T\mathbf{U}_l = I. \tag{29}$$

Here, $\mathbf{U}_l$ is the $g \times g$ orthogonal optimization variable, and $(\cdot)_{:m,:m}$ denotes the top-left $m \times m$ submatrix. The solution to this trace maximization problem is obtained by performing an eigendecomposition on the summed covariance matrix $\sigma_{x,l} + \sigma_{y,l}$. The resulting matrix $\mathbf{U}_l$, whose columns are the eigenvectors sorted in descending order of their corresponding eigenvalues, is the optimal orthogonal transformation $\mathbf{U}_l$.

By applying this rotation, we ensure that the principal components from both the real and imaginary dimensions of the keys are aligned and concentrated within the first few heads. Consequently, we can discard the RoPE components from the remaining heads in both queries and keys while preserving the most significant positional information, thereby minimizing the performance degradation.

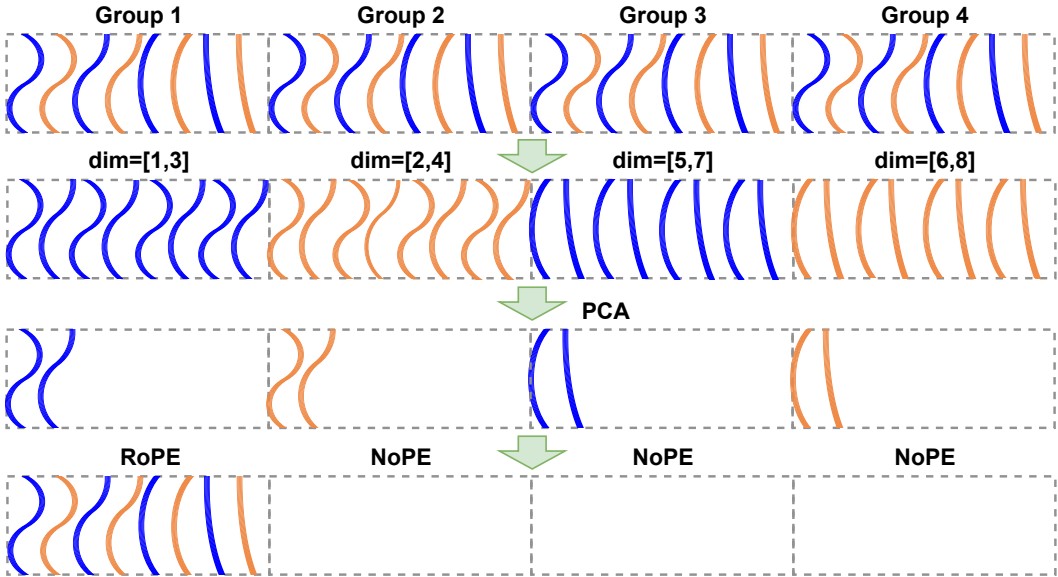

Figure 7: Pipeline of RoRoPE with FreqFold. RoRoPE encodes the entire frequency spectrum of all attention heads in a *single* latent dimension, which limits its expressive power. FreqFold remedies this by clustering adjacent-frequency dimensions and extracting their principal components jointly, allocating a higher-dimensional subspace to similar features. This richer representation enables $K_{rope}$ to retain far more positional information.

## D FreqFold: Detailed Mechanism, Example, and PCA Efficiency

This appendix provides a detailed explanation of the FreqFold technique, illustrates its operation with a concrete example, and formally connects its benefits to a general principle of Principal Component Analysis (PCA) concerning structured data. This justification clarifies FreqFold's role in minimizing transformation loss towards decoupled RoPE within the RoRoPE framework (Section 4.2).

### D.1 Detailed Explanation of FreqFold and RoRoPE's PCA

In the RoRoPE framework, Rotary Position Embedding (RoPE) is applied. RoPE encodes positional information by rotating pairs of feature dimensions. For each RoPE frequency index $l \in \{1, \ldots, d/2\}$, the corresponding pair of dimensions $([2l-1 :: d], [2l :: d])$ from key vectors are rotated. When multiple original attention heads are used (say, $g$ heads), and their key/query projection outputs are concatenated, the RoPE operation for a specific frequency index $l$ applies to a $2g$-dimensional vector segment (formed by concatenating the $l$-th 2D RoPE subspace from each of the $g$ heads). RoRoPE then applies PCA via matrices $\{\mathbf{U}_l\}_{l=1}^{d/2}$ to these $2g$-dimensional segments, independently for each frequency index $l$.

The core idea of FreqFold is to approximate numerically similar RoPE base frequencies as being effectively identical. For instance, if RoPE uses original base frequencies $\theta_{l_1}, \theta_{l_2}, \ldots, \theta_{l_M}$ that are close in value, $M$D-FreqFold might treat them all as a single, representative frequency $\theta^*$.

This approximation has a significant implication for how PCA is applied in RoRoPE:

- **Without FreqFold (Standard RoRoPE PCA):** For each distinct RoPE frequency index $l$, a separate PCA transformation $\mathbf{U}_l$ is learned and applied to the corresponding $2g$-dimensional key/query segments.

- **With FreqFold:** If $M$ original RoPE frequency indices (say $l_1, \ldots, l_M$) are grouped together by FreqFold due to their frequency similarity, the $M$ corresponding $2g$-dimensional segments are effectively concatenated. Instead of $M$ separate PCAs on $2g$-dimensional vectors, a single PCA is performed on the resulting $M \cdot 2g$-dimensional vectors.

### D.1.1 Illustrative Example of FreqFold

Let's consider a scenario with $g = 2$ key heads, and each head has $d_{head} = 8$ dimensions. Thus, there are $d/2 = 8/2 = 4$ distinct RoPE frequency indices per head, which we denote as $\phi_1, \phi_2, \phi_3, \phi_4$. The total number of dimensions is $2 \times 8 = 16$. The RoPE angles for these 16 dimensions could be conceptualized as follows (repeating for each pair, and across heads):

- **Head 1 (dims 1-8):** $(\phi_1, \phi_1), (\phi_2, \phi_2), (\phi_3, \phi_3), (\phi_4, \phi_4)$
- **Head 2 (dims 9-16):** $(\phi_1, \phi_1), (\phi_2, \phi_2), (\phi_3, \phi_3), (\phi_4, \phi_4)$

**Case 1: RoRoPE without FreqFold** For each frequency index $\phi_l$, RoRoPE groups the corresponding dimensions from all $g = 2$ heads. Each such group forms $2g = 2 \times 2 = 4$-dimensional vectors (across $N$ samples).

- Group for $\phi_1$: Dimensions $\{1, 2\}$ from Head 1 and $\{9, 10\}$ from Head 2. PCA is applied to these $N$ samples of 4D vectors.
- Group for $\phi_2$: Dimensions $\{3, 4\}$ from Head 1 and $\{11, 12\}$ from Head 2. PCA is applied to these $N$ samples of 4D vectors.
- Group for $\phi_3$: Dimensions $\{5, 6\}$ from Head 1 and $\{13, 14\}$ from Head 2. PCA is applied to these $N$ samples of 4D vectors.
- Group for $\phi_4$: Dimensions $\{7, 8\}$ from Head 1 and $\{15, 16\}$ from Head 2. PCA is applied to these $N$ samples of 4D vectors.

Here, RoRoPE performs 4 separate PCA operations.

**Case 2: RoRoPE with 2D-FreqFold** 2D-FreqFold implies we are pairing up original frequencies. Suppose FreqFold approximates $\phi_1 \approx \phi_2$ (calling this effective frequency $\Phi_A = \phi_1$) and $\phi_3 \approx \phi_4$ (calling this $\Phi_B = \phi_3$).

- **Effective Group for $\Phi_A$:** This group now includes all dimensions originally associated with $\phi_1$ OR $\phi_2$.
    - Original $\phi_1$-dimensions: $\{1, 2\}$ from Head 1; $\{9, 10\}$ from Head 2. (Forms a 4D segment $S_{\phi_1}$)
    - Original $\phi_2$-dimensions: $\{3, 4\}$ from Head 1; $\{11, 12\}$ from Head 2. (Forms a 4D segment $S_{\phi_2}$)

    With FreqFold, these segments $S_{\phi_1}$ and $S_{\phi_2}$ are concatenated. PCA is now applied to the $N$ samples of $(4 + 4) = 8$-dimensional vectors formed by $[S_{\phi_1}, S_{\phi_2}]$. Effectively, dimensions $\{1, 2, 3, 4\}$ from Head 1 are combined with $\{9, 10, 11, 12\}$ from Head 2.

- **Effective Group for $\Phi_B$:** Similarly, this group includes dimensions originally for $\phi_3$ OR $\phi_4$.
    - Original $\phi_3$-dimensions: $\{5, 6\}$ from Head 1; $\{13, 14\}$ from Head 2. (Forms $S_{\phi_3}$)
    - Original $\phi_4$-dimensions: $\{7, 8\}$ from Head 1; $\{15, 16\}$ from Head 2. (Forms $S_{\phi_4}$)

    PCA is applied to the $N$ samples of 8-dimensional vectors formed by $[S_{\phi_3}, S_{\phi_4}]$.

Here, RoRoPE with FreqFold performs 2 PCA operations, but each operates on larger, 8-dimensional vectors which are concatenations of what were previously separate PCA targets.

### D.2 Formalizing the Benefit of FreqFold in PCA

The example above illustrates that FreqFold causes a re-grouping and concatenation of data segments prior to PCA. The benefit of this concatenation is explained by the following proposition. It states that performing PCA jointly on these concatenated segments (as FreqFold enables) is more effective at preserving variance (and thus minimizing loss) than the alternative of performing separate PCAs on the original, smaller segments and then notionally combining their outcomes.

Consider one such FreqFold merge: suppose $M$ original RoPE frequency indices $l_1, \ldots, l_M$ are deemed equivalent by FreqFold. Without FreqFold, each $l_p$ would correspond to a dataset $X_p$ (e.g., $N$ samples of $2g$-dimensional key segments). With FreqFold, these $M$ datasets are concatenated into a single larger dataset $X_{merged} = [X_1, X_2, \ldots, X_M]$, and PCA is applied to $X_{merged}$.

**Proposition 2.** *Let $M$ distinct groups of key segments $X_1, X_2, \ldots, X_M$ be identified. Each $X_p \in \mathbb{R}^{N \times d'}$ (where $p \in \{1, \ldots, M\}$) consists of $N$ samples of $d'$-dimensional vectors. Assume data in each $X_p$ is mean-centered. Let $S_p = \frac{1}{N-1} X_p^T X_p \in \mathbb{R}^{d' \times d'}$ be its covariance matrix. FreqFold causes these $M$ groups to be merged for a single PCA operation.*

*Define $V_1 = \sum_{p=1}^{M} \lambda_{p,1}$, where $\lambda_{p,1}$ is the largest eigenvalue of $S_p$. This $V_1$ represents the sum of variances if each of the $M$ original groups $X_p$ were individually reduced to its single most dominant dimension.*

*Let $Z = [X_1, X_2, \ldots, X_M] \in \mathbb{R}^{N \times (M \cdot d')}$ be the dataset formed by concatenating the features (columns) of these $M$ groups. Let $S_{concat} = \frac{1}{N-1} Z^T Z \in \mathbb{R}^{(M \cdot d') \times (M \cdot d')}$ be its covariance matrix. Define $V_2 = \sum_{j=1}^{M} \mu_j$, where $\mu_1 \geq \mu_2 \geq \ldots \geq \mu_M$ are the $M$ largest eigenvalues of $S_{concat}$. This $V_2$ represents the variance captured if the concatenated data $Z$ is reduced to $M$ dimensions using PCA.*

*Then, the variance captured by the joint PCA on the FreqFold-merged data ($V_2$) is greater than or equal to the sum of variances from optimally reducing each original group to one dimension ($V_1$):*

$$V_2 \geq V_1$$

This proposition explains that FreqFold's strategy of enabling PCA over larger, concatenated segments (formed by merging data from RoPE frequencies deemed similar) is mathematically favored for variance preservation compared to separate, more fragmented PCAs.

### D.3 Proof of Proposition 2

The objective is to prove that $V_2 \geq V_1$, using the notation from Proposition 2. The proof strategy is to construct a specific $M$-dimensional subspace for the concatenated data $Z$. We show that the variance captured by projecting $Z$ onto this particular subspace equals $V_1$. Since the PCA procedure yielding $V_2$ finds the optimal $M$-dimensional subspace maximizing captured variance, $V_2$ must be at least $V_1$.

Let $\lambda_{p,1}$ be the largest eigenvalue of $S_p$ (covariance of $X_p$), and $\boldsymbol{w}_{p,1} \in \mathbb{R}^{d'}$ be its corresponding eigenvector. So, $S_p \boldsymbol{w}_{p,1} = \lambda_{p,1} \boldsymbol{w}_{p,1}$ and $\boldsymbol{w}_{p,1}^T \boldsymbol{w}_{p,1} = 1$. The variance $\lambda_{p,1} = \boldsymbol{w}_{p,1}^T S_p \boldsymbol{w}_{p,1}$. $V_1 = \sum_{p=1}^{M} \lambda_{p,1}$.

For the concatenated data $Z$, $V_2 = \sum_{j=1}^{M} \mu_j$. By Ky Fan's theorem for matrix eigenvalues:

$$V_2 = \max_{\substack{U \in \mathbb{R}^{(M \cdot d') \times M} \\ U^T U = I_M}} \text{Tr}(U^T S_{concat} U)$$

where $U$'s columns form an orthonormal basis for an $M$-dimensional subspace of $\mathbb{R}^{M \cdot d'}$.

Construct $U^* = [\boldsymbol{u}_1^*, \ldots, \boldsymbol{u}_M^*] \in \mathbb{R}^{(M \cdot d') \times M}$. For $p \in \{1, \ldots, M\}$, define $\boldsymbol{u}_p^* \in \mathbb{R}^{M \cdot d'}$:

$$\boldsymbol{u}_p^* = \begin{pmatrix} \mathbf{0}_{d' \times 1} \\ \vdots \\ \boldsymbol{w}_{p,1} \quad \text{(as the $p$-th block of size $d'$)} \\ \vdots \\ \mathbf{0}_{d' \times 1} \end{pmatrix}$$

The set $\{\boldsymbol{u}_1^*, \ldots, \boldsymbol{u}_M^*\}$ is orthonormal. The variance retained by projecting $Z$ onto the subspace of $U^*$ is:

$$\text{Tr}((U^*)^T S_{concat} U^*) = \sum_{p=1}^{M} (\boldsymbol{u}_p^*)^T S_{concat} \boldsymbol{u}_p^*$$

Let $S_{qr}$ be the $(q,r)$-th block of $S_{concat}$, where $S_{qr} = \frac{1}{N-1} X_q^T X_r$. Note $S_{pp} = S_p$. Each term $(\boldsymbol{u}_p^*)^T S_{concat} \boldsymbol{u}_p^* = \boldsymbol{w}_{p,1}^T S_{pp} \boldsymbol{w}_{p,1} = \boldsymbol{w}_{p,1}^T S_p \boldsymbol{w}_{p,1} = \lambda_{p,1}$. So, $\text{Tr}((U^*)^T S_{concat} U^*) = \sum_{p=1}^{M} \lambda_{p,1} = V_1$. Since $V_2$ is the maximum possible variance:

$$V_2 \geq \text{Tr}((U^*)^T S_{concat} U^*) = V_1$$

Thus, $V_2 \geq V_1$. This proves Proposition 2.

## D.4 Discussion on the Trade-off in FreqFold

While Proposition 2 demonstrates a clear benefit of FreqFold in terms of PCA efficiency—specifically, that merging M original frequency groups allows for greater variance preservation when reducing to M dimensions—it is crucial to acknowledge an inherent trade-off. The foundational assumption of FreqFold is the approximation of numerically similar RoPE base frequencies as effectively identical. This approximation, by its very nature, introduces a degree of deviation from the original, precise RoPE formulation.

The extent of this deviation, and thus the potential loss in the fidelity of positional encoding, typically correlates with how aggressively frequencies are grouped. A larger $M$ or a looser criterion for similarity when grouping frequencies can amplify this approximation error. Consequently, while increasing the dimensionality of vectors undergoing PCA is beneficial from the perspective of PCA variance capture as shown by the proposition, it may simultaneously increase the lossiness of the RoPE approximation itself. Therefore, the practical application of FreqFold requires a careful balancing act. The parameter M (representing the number of original RoPE frequencies treated as one effective frequency for PCA purposes) or the specific grouping strategy for frequencies must be chosen to optimize this trade-off.

# E   Balancing Key-Value Norms and Low-Rank Approximation

This appendix elaborates on the Key-Value (KV) balancing technique and the subsequent joint low-rank approximation applied to the NoPE (No Positional Encoding) components of the keys and the values, as mentioned in Section 4.3 of the main paper. After the RoRoPE procedure (Section 4.2), the key projection matrix $W^K$ is effectively split into two components: $W^{DK}_{\text{rope}} \in \mathbb{R}^{d \times D}$ corresponding to the single head that retains RoPE, and $W^{DK}_{\text{nope}} \in \mathbb{R}^{(g-1)d \times D}$ corresponding to the remaining $g-1$ head components that do not use RoPE. The value projection matrix is denoted as $W^{DV} \in \mathbb{R}^{gd \times D}$.

## E.1   KV Balancing: Purpose and Formulation

**Purpose**   The primary goal of KV balancing is to ensure that the principal component analysis (PCA), when applied jointly to the NoPE key and value activations, is not disproportionately influenced by components with larger norms. We observed that the activations derived from $W^{DK}_{\text{nope}}$ (i.e., $\mathbf{k}_{\mathbf{NoPE},t} = W^{DK}_{\text{nope}} \mathbf{x}_t$) often have a significantly larger average norm than those from $W^{DV}$ (i.e., $\mathbf{v}_t = W^{DV} \mathbf{x}_t$). Without balancing, PCA would predominantly capture the variance within the NoPE key components, potentially neglecting important variations in the value components.

**Formulation**   To address this imbalance, we introduce a scaling factor $\alpha$. This factor is computed as the ratio of the expected L2 norms of the NoPE key activations to the value activations, based on a calibration dataset:

$$\alpha = \frac{\mathbb{E}_t[\|W^{DK}_{\text{nope}} \mathbf{x}_t\|_2]}{\mathbb{E}_t[\|W^{DV} \mathbf{x}_t\|_2]} \tag{30}$$

where $\mathbf{x}_t \in \mathbb{R}^D$ is the $t$-th input token.

While the main paper states scaling $W^{DK}_{\text{nope}}$ by $1/\alpha$ and $W^{UK}$ by $\alpha$ for mathematical equivalence in the model's output, for the purpose of deriving the PCA projection, we effectively use scaled NoPE key activations. That is, the activations used to compute the PCA basis are $\mathbf{k}'_{\text{NoPE},t} = 1/\alpha \cdot W^{DK}_{\text{nope}} \mathbf{x}_t$ and $\mathbf{v}_t = W^{DV} \mathbf{x}_t$. This ensures that the PCA process considers features from keys and values on a more equitable footing with respect to their magnitudes. The subsequent low-rank decomposition will then be applied to $W^{DK}_{\text{nope}}$ and $W^{DV}$, using the PCA basis derived from these balanced activations.

## E.2   Joint Low-Rank Approximation of NoPE Keys and Values using PCA

After determining the scaling factor $\alpha$, we proceed to compress the projection matrices associated with the NoPE keys ($W^{DK}_{\text{nope}}$) and all values ($W^{DV}$) jointly.

The process is as follows:

1. **Collect Calibrated Activations**: A small calibration dataset (WikiText-2) is used. For each input $\mathbf{x}_t$ from this dataset, we compute the scaled NoPE key activations $\mathbf{k}'_{\text{NoPE},t}$ and the value activations $\mathbf{v}_t$. These are concatenated to form combined activation vectors:

$$\mathbf{c}_{\text{NoPE},t} = \begin{pmatrix} \mathbf{k}'_{\text{NoPE},t} \\ \mathbf{v}_t \end{pmatrix} \in \mathbb{R}^{(2g-1)d} \tag{31}$$

2. **Perform PCA**: PCA is performed on the set of collected combined activation vectors $\{\mathbf{c}_{\text{NoPE},t}\}$. This involves computing the covariance matrix of these vectors and finding its principal components. The eigenvectors (corresponding to the largest eigenvalues) are selected to form the columns of a projection matrix $R_{KV} \in \mathbb{R}^{((2g-1)d) \times r_{kv}}$, where $r_{kv}$ is the reduced rank. This matrix $R_{KV}$ captures the directions of highest variance in the (balanced) combined NoPE key and value activation space.

3. **Low-Rank Decomposition of Projection Matrices**: Let $W^{DKV} = \begin{pmatrix} W^{DK}_{\text{nope}} \\ W^{DV} \end{pmatrix} \in \mathbb{R}^{((2g-1)d) \times D}$ be the initial projection matrix that transforms the input $\mathbf{x}_t$ into an intermediate NoPE Key and Value representation $\mathbf{c}_{\text{NoPE},t} = W^{DKV}\mathbf{x}_t$. Further, let $W^{UKV} = \begin{pmatrix} W^{UK}_{\text{nope}} & 0 \\ 0 & W^{UV} \end{pmatrix} \in \mathbb{R}^{2hd \times ((2g-1)d)}$ represent the subsequent collective projection matrix that takes $\mathbf{c}_{\text{NoPE},t}$ and processes it to produce the actual keys and values required by the attention mechanism for the NoPE components, where $W^{UK}_{\text{rope}} \in \mathbb{R}^{hd \times gd}$ and $W^{UK}_{\text{nope}} \in \mathbb{R}^{hd \times (g-1)d}$ are two parts of $W^{UK}$ hat participate in and do not participate in the RoPE computation, respectively. The original sequence of operations for these components can be expressed as $W^{UKV}W^{DKV}\mathbf{x}_t \in \mathbb{R}^{2hd}$, in which the first $hd$ elements correspond to the keys and the following $hd$ elements correspond to the values.

To introduce a low-rank bottleneck, we modify both $W^{DKV}$ and $W^{UKV}$ using the PCA projection matrix $R_{KV}$.

- The initial projection matrix $W^{DKV}$ is transformed into $W^{DKV'} \in \mathbb{R}^{r_{kv} \times D}$:

$$W^{DKV'} = R^T_{KV}W^{DKV} \tag{32}$$

  This new matrix $W^{DKV'}$ takes the original input $\mathbf{x}_t$ and projects it into a compressed $r_{kv}$-dimensional latent space, which is the actual content stored in the KV cache for the NoPE components.

- The subsequent projection matrix $W^{UKV}$ is transformed into $W^{UKV'} \in \mathbb{R}^{2hd \times r_{kv}}$:

$$W^{UKV'} = W^{UKV}R_{KV} \tag{33}$$

  This new matrix $W^{UKV'}$ now takes the compressed latent representation as input and produces the final representations for the NoPE components that are used in the attention calculation. As we can see, $W^{UKV'}$ is actually the concatenated form of $W^{UK}$ and $W^{UV}$ in MLA:

$$W^{UKV'} = \begin{pmatrix} W^{UK} \\ W^{UV} \end{pmatrix} \tag{34}$$

This joint decomposition allows for a more holistic compression by identifying shared latent structures between NoPE keys and values, guided by the balanced PCA.

## F  Theoretical Analysis of Balancing Scheme for PCA in Matrix Decomposition

### F.1  Theoretical Analysis

In this section, we focus on a situation where two matrices, $W_k$ and $W_v$, are concatenated together for dimensionality reduction. A potential issue arises when the matrices have different magnitudes, which leads to an imbalance in the relative errors during reconstruction. The relative error for one matrix may be small while for the other it may be large. To address this, we propose scaling both matrices by their Root Mean Square (RMS) before concatenating. This scaling ensures that the relative errors for both matrices are approximately equal. The goal of this section is to theoretically prove the validity of this approach by formulating a loss function that reflects the relative errors and demonstrating that the scaling values we mentioned will minimize this loss.

### F.1.1 Problem Setup

We are given two matrices $W_k$ and $W_v$ of dimensions $m \times n_k$ and $m \times n_v$, respectively. The matrices are concatenated as follows:

$$W = \left[ \frac{W_k}{a}, \frac{W_v}{b} \right] \tag{35}$$

where $a$ and $b$ are scaling factors that need to be optimized. These scaling factors are introduced to balance the relative errors during reconstruction. We perform PCA on the concatenated matrix $W$. The SVD of the matrix $W$ is given by:

$$W = U\Sigma V^T = \sum_{i=1}^{rank(W)} u_i \sigma_i v_i^T \tag{36}$$

where $U$, $\Sigma$, and $V$ are singular vectors, singular values, and right singular vectors, respectively. The $r$-rank approximation of matrix $W$ is given by:

$$W' = U_r \Sigma_r V_r^T = \sum_{i=1}^{r} u_i \sigma_i v_i^T \tag{37}$$

$$W' = \left[ \frac{W_k'}{a}, \frac{W_v'}{b} \right] \tag{38}$$

where $W_k'$ and $W_v'$ are the low-rank approximations of $W_k$ and $W_v$. When the scaling factors $a = 1$ and $b = 1$, the matrices $W_k$ and $W_v$ are not scaled and directly undergo low-rank approximation. The matrices $W_k'$ and $W_v'$ represent the low-rank approximations of $W_k$ and $W_v$ without the effect of any relative scaling based on the Frobenius norms.

To address the imbalance in the relative errors during reconstruction, we define the loss function to equally quantify the relative errors between the original and the approximated matrices. The loss is the sum of the relative errors for $W_k$ and $W_v$, given by:

$$\mathcal{L}(a, b) = \| \frac{W_K - W_K'}{RMS(W_K)} \|_F^2 + \| \frac{W_V - W_V'}{RMS(W_V)} \|_F^2 \tag{39}$$

Next, we will derive the condition when the loss function reaches its minimum, specifically when the scaling factors $a$ and $b$ satisfy the ratio:

$$\frac{a}{b} = \frac{\text{RMS}(W_k)}{\text{RMS}(W_v)} \tag{40}$$

### F.1.2 Proof of Proportionality

Before proceeding with the derivation, we first need to verify the proportionality between $a$ and $b$, which implies that the absolute values of $a$ and $b$ do not affect the conclusion. As long as the ratio $a : b$ is fixed, the final loss will be the same, regardless of the absolute values of $a$ and $b$.

Consider scaling the factors $a$ and $b$ by a constant $c > 0$: $a' = ca, \quad b' = cb$.

The concatenated matrix with the new scaling factors is:

$$W_{\text{new}} = \left[ \frac{W_k}{a'}, \frac{W_v}{b'} \right] = \left[ \frac{W_k}{ca}, \frac{W_v}{cb} \right] = \frac{1}{c} W = \frac{1}{c} U\Sigma V^T \tag{41}$$

Thus, the SVD of $W_{\text{new}}$ has the same left and right singular vectors $U$ and $V$ as $W$, but its singular values are scaled by $\frac{1}{c}$. The low-rank approximation of $W_{\text{new}}$ is:

$$W_{\text{new}}' = U_r \left( \frac{1}{c} \Sigma_r \right) V_r^T = \frac{1}{c} \left( U_r \Sigma_r V_r^T \right) = \frac{1}{c} W'. \tag{42}$$

Comparing this with the block definition of $W_{\text{new}}'$, we identify:

$$W_{k,\text{new}}' = W_k', \quad W_{v,\text{new}}' = W_v'. \tag{43}$$

The loss function $\mathcal{L}(a, b)$ is invariant under scaling $a \mapsto ca$ and $b \mapsto cb$ for any $c > 0$. Thus, it depends only on the ratio $\frac{a}{b}$. Therefore, in the following derivations, we will always assume that $b = 1$.

### F.1.3 Verify Minimum Conditions

We now verify the minimum condition when $a = \frac{\text{RMS}(W_k)}{\text{RMS}(W_v)}$. For convenience, we define $\frac{W_k}{a} = A$, $\frac{W_v}{b} = B$, $\frac{W_k'}{a} = C$, $\frac{W_v'}{b} = D$.

$$\mathcal{L}(a, 1) = \frac{\|A - C\|_F^2}{\|A\|_F^2} mn_k + \frac{\|B - D\|_F^2}{\|B\|_F^2} mn_v \tag{44}$$

$$\frac{d\mathcal{L}(a, 1)}{da} = \frac{d}{da}\left( \frac{\|A - C\|_F^2}{\|A\|_F^2} mn_k + \frac{\|B - D\|_F^2}{\|B\|_F^2} mn_v \right) \tag{45}$$

$$= \|A - C\|_F^2 \frac{d}{da}\left( \frac{1}{\|A\|_F^2} \right) mn_k + \frac{mn_k \frac{d}{da}\left( \|A - C\|_F^2 \right)}{\|A\|_F^2} \tag{46}$$

$$+ \frac{mn_v \frac{d}{da}\left( \|B - D\|_F^2 \right)}{\|B\|_F^2} \tag{47}$$

when $b = 1, a = \frac{\text{RMS}(W_k)}{\text{RMS}(W_v)}$,

$$\frac{d\mathcal{L}(a, 1)}{da} = \frac{2\|A - C\|_F^2}{a\|A\|_F^2} mn_k + \frac{mn_v \frac{d}{da}\left( \|A - C\|_F^2 + \|B - D\|_F^2 \right)}{\|B\|_F^2} \tag{48}$$

$$= \frac{2\|A - C\|_F^2}{a\|A\|_F^2} mn_k + \frac{mn_v \frac{d}{da}\left( \|W - W'\|_F^2 \right)}{\|B\|_F^2} \tag{49}$$

$$= \frac{2\|A - C\|_F^2}{a\|A\|_F^2} mn_k + \frac{mn_v \frac{d}{da}\left( \sum_{i=r+1}^{rank(W)} \sigma_i^2 \right)}{\|B\|_F^2} \tag{50}$$

$$\tag{51}$$

$$W = U\Sigma V^T \tag{52}$$
$$dW = Ud\Sigma V^T + U\Sigma dV^T + dU\Sigma V^T \tag{53}$$
$$U^T dW V = d\Sigma + \Sigma dV^T V + U^T dU\Sigma V^T \tag{54}$$

Since $d(U^T U) = dU^T U + U^T dU = 0$ and $d(V^T V) = dV^T V + V^T dV = 0$, $U^T dU$ and $V^T dV$ are skew-symmetric matrices, and the diagonal elements are zero. therefore,

$$\frac{d\sigma_i}{da} = u_i^T \frac{dW}{da} v_i = u_i^T [-\frac{A}{a}, 0]v_i \tag{55}$$

$$\frac{d\mathcal{L}(a, 1)}{da} = \frac{2\|A - C\|_F^2}{a\|A\|_F^2} mn_k + \frac{\sum_{i=r+1}^{rank(W)} 2mn_v \sigma_i \frac{d\sigma_i}{da}}{\|B\|_F^2} \tag{56}$$

$$= \frac{2\|A - C\|_F^2}{a\|A\|_F^2} mn_k - \frac{\sum_{i=r+1}^{rank(W)} 2mn_v \sigma_i u_i^T [A, 0]v_i}{a\|B\|_F^2} \tag{57}$$

$$A = W[:, : n_k] = \sum_{i=1}^{rank(W)} u_i \sigma_i v_i^T[: n_k] \tag{58}$$

$$C = W'[:, : n_k] = \sum_{i=1}^{r} u_i \sigma_i v_i^T[: n_k] \tag{59}$$

$$A - C = W'[:, : n_k] = \sum_{i=r+1}^{rank(W)} u_i \sigma_i v_i^T[: n_k] \tag{60}$$

therefore,

$$\|A - C\|_F^2 = Tr((A - C)^T(A - C)) = \sum_{i=r+1}^{rank(W)} \sigma_i^2 v_i^T[: n_k]v_i[: n_k] \tag{61}$$

Table 2: Composition of the training dataset.

| Dataset | Sampling Weight |
|---|---|
| fineweb-edu-dedup | 0.70 |
| cosmopedia-v2 | 0.15 |
| python-edu | 0.06 |
| open-web-math | 0.08 |
| stackoverflow | 0.01 |

$$\sigma_i u_i^T [A, 0] v_i = \sigma_i^2 v_i^T[: n_k] v_i[: n_k] \tag{62}$$

when $b = 1, a = \frac{\text{rms}(W_k)}{\text{rms}(W_v)}$, we have:

$$\frac{d\mathcal{L}(a, 1)}{da} = 0 \tag{63}$$

Thus far, we have demonstrated that when the loss function is

$$\mathcal{L}(a, b) = \left\| \frac{W_K - W_K'}{\text{RMS}(W_K)} \right\|_F^2 + \left\| \frac{W_V - W_V'}{\text{RMS}(W_V)} \right\|_F^2 \tag{64}$$

the ratio

$$\frac{a}{b} = \frac{\text{RMS}(W_k)}{\text{RMS}(W_v)} \tag{65}$$

achieves the best compression performance.

# G   Experimental Settings of Fine-tuning

**Datasets**   Following the experimental setups of MHA2MLA, we fine-tune our models using the prtraining corpus from SmolLM [40]. The dataset comprises FineWeb-Edu-Dedup [41], Cosmopedia-v2 — a synthetic dataset generated by Mixtral [42], Python-Edu from StarCoder [43], Open-Web-Math [44], and data from StackOverflow [45]. To ensure a fair comparison with the MHA2MLA baseline, we constructed our training dataset using the same data composition strategy. Specifically, we replicate the dataset mixing ratios used in the MHA2MLA setup to maintain experimental consistency, which is shown in Table 2.

**Hyperparameters**   The fine-tuning hyperparameters for models of all sizes are listed in Table 3. In the table, entries with a slash (/) indicate a two-step training process.

Table 3: Training details across different models.

| | SmolLM 1B7 | | LLaMA2 7B | | |
|---|---|---|---|---|---|
| | -68.75% | -87.50% | -68.75% | -87.50% | -92.97% |
| Batch size | 64 | 64 | 64 | 64 / 64 | 256 / 64 |
| Learning rate | 1e-4 | 1e-4 | 2e-5 | 2e-5 / 2e-5 | 1e-4 / 2e-5 |
| Tokens | 300M | 1B | 500M | 2B / 1B | 5B / 1B |
| Warmup ratio | 0.03 | 0.08 | 0 | 0 / 0.03 | 0 / 0.03 |
| lr scheduler | constant | constant | constant | constant / cosine | constant / cosine |
| Sequence length | 2048 | 2048 | 4096 | 4096 | 4096 |

# H   vLLM Benchmark Details

In Section 5.4, we demonstrated the speedup achieved by TransMLA—which compresses 92.97% of the KV cache—compared to the original LLaMA-2-7B model. This section provides the detailed methodology and hardware configurations for this benchmark.

To account for the effects of both the prefilling and decoding stages, we adopt a setting where the input and output lengths are equal. For instance, with a total context length of 1k, we set the input

length to 512 tokens and the output length to 512 tokens. Most experiments are conducted using 100 requests to compute the average throughput. However, for shorter context lengths such as 1k, inference is extremely fast, leading to some timing fluctuations. To mitigate this, we increase the number of requests to 1000 for more stable measurements.

While the original LLaMA-2-7B model supports a maximum context length of 4096 tokens, we extend this limit to 32k tokens in our evaluation. Detailed throughput results are presented in Table 4.

On a GPU with 165.2 TFLOPS of compute and 24GB of memory, the LLaMA-2-7B model runs out of memory when the context length reaches 16k tokens. In contrast, TransMLA sustains a throughput of 414.41 tokens per second under the same conditions. On a more powerful GPU with 320 TFLOPS and 64GB of memory, we employ a development version of the vLLM framework. We anticipate that the throughput of TransMLA will improve further with the release of future optimized versions of the framework tailored for this hardware.

Table 4: Throughput comparison between LLaMA-2-7b and TransMLA at varying input lengths and number of requests.

| Context Length | Requests | Model | Throughput(output tokens/s) | | |
|---|---|---|---|---|---|
| | | | 165.2 TF\|24GB | 312 TF\|40GB | 320 TF\|64GB |
| 1K | 1000 | LLaMA-2-7b | 653.81 | 1579.26 | 1249.13 |
| | | TransMLA | **3043.65** | **4062.43** | **1798.17** |
| 2K | 100 | LLaMA-2-7b | 352.85 | 850.14 | 789.31 |
| | | TransMLA | **2241.87** | **2577.01** | **1080.73** |
| 4K | 100 | LLaMA-2-7b | 173.09 | 441.37 | 442.63 |
| | | TransMLA | **1318.78** | **1926.15** | **1021.03** |
| 8K | 100 | LLaMA-2-7b | 85.80 | 218.51 | 216.66 |
| | | TransMLA | **832.69** | **1118.18** | **870.15** |
| 16K | 100 | LLaMA-2-7b | OOM | 110.58 | 112.13 |
| | | TransMLA | **414.41** | **601.36** | **483.22** |
| 32K | 100 | LLaMA-2-7b | OOM | 38.32 | 55.69 |
| | | TransMLA | OOM | **243.81** | **278.09** |

# I    Inference Speed Ablation

The strength of the MLA design extends beyond its use of low-rank compression—a concept that has been explored in prior works. Its true innovation lies in the Absorb operation, which plays a pivotal role in enhancing efficiency. During the training phase (which is compute-intensive), this operation enables attention computations to be performed using compact 192-dimensional representations per head, resulting in high computational efficiency. In the inference phase (which is memory-bound), it utilizes a shared 576-dimensional KV cache across heads by absorbing the key projections into the query projection matrices. This mechanism significantly reduces memory access overhead, both on high-bandwidth memory (HBM) and on-chip SRAM.

Table 5: Comparison of the impact of `num_kv_heads`, `head_dim`, and parameters on inference speed for GQA and MLA. To eliminate the influence of MLP layers on the results, the speed test was conducted using models composed of 32 attention layers only.

| Method | $h_{kv}$ | $d$ | KV Cache | Param (M) | Throughput (tokens/s) |
|---|---|---|---|---|---|
| MLA (with Absorb) | 1*1 | 576 | 576 | 48.5 | 4151.17 |
| MLA (without Absorb) | 32*2 | 192 | 12288 | 48.5 | 248.85 |
| GQA | 4*2 | 128 | 1024 | 37.7 | 2714.80 |
| | 8*2 | 128 | 2048 | 41.9 | 1436.92 |
| | 16*2 | 64 | 2048 | 41.9 | 1118.83 |

As shown in Table 5 (rows 2–3), disabling the Absorb operation leads to a 16.7× degradation in MLA's inference speed at a 16k context length, underscoring its crucial role in achieving efficiency.

This highlights the necessity of supporting the Absorb operation, as it is indispensable for realizing the practical benefits of MLA in real-world deployment. The importance of the Absorb operation is further corroborated by findings from Palu [20]. In their Figure 5, they demonstrate that removing RoPE allows the Absorb operation to deliver a 6.17× speedup at a 64k context length. However, when RoPE is retained, Absorb must be disabled, reducing the speedup to 2.91×. Unfortunately, since Palu does not support RoPE decoupling, it cannot apply the Absorb operation to widely used architectures such as LLaMA, Qwen, and Mistral.

In Figure 5, we only present the inference speed of the LLaMA-2 model based on MHA. Since GQA already requires fewer KV cache entries than MHA, it is natural to ask whether TransMLA can further improve the inference speed of GQA-based models. To investigate this, we conducted experiments using LLaMA-3-8B, whose key and value projections each contain 8 heads with 128 dimensions per head, resulting in a total KV cache size of 2048 dimensions. We applied TransMLA to reduce the KV cache to 576 dimensions and measured the inference speed across different GPUs. As shown in Figure 8, TransMLA still achieves approximately a 3× speedup, demonstrating its effectiveness even when applied to GQA architectures.

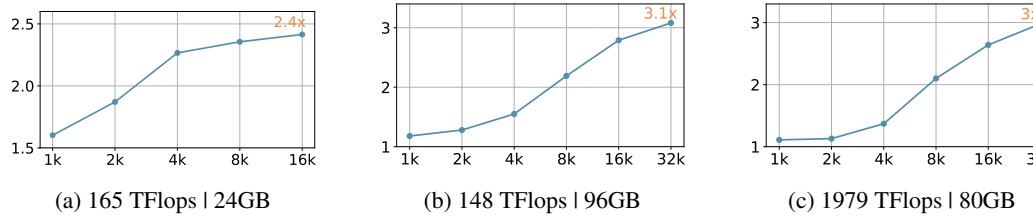

(a) 165 TFlops | 24GB          (b) 148 TFlops | 96GB          (c) 1979 TFlops | 80GB

Figure 8: Inference speedups with TransMLA compared to the original LLaMA3-8B model on three AI accelerators. **Context length** represents the total sequence length.

## J  Evaluation on GQA Models

The goal of Table 1 is to provide a fair and consistent comparison between TransMLA and prior relevant work. Therefore, we selected the recently published MHA2MLA [23] as our baseline, as it also focuses on RoPE decoupling and KV cache compression. Following their experimental setup, we used two MHA-based models—SmolLM-1.7B and LLaMA-2-7B. In fact, TransMLA is fully compatible with GQA. As stated in related work, both MHA and MQA can be regarded as special cases of GQA. Specifically, when the number of query heads equals the number of key-value heads in GQA, it effectively reduces to an MHA. In this section, we further extend our evaluation to several mainstream GQA models, and the corresponding results without training are presented in Table 6.

Table 6: Evaluated with Wikitext-2 Perplexity (PPL) of Converting GQA Models to MLA (lower is better). Original PPL refers to the perplexity of the original GQA model without conversion. Decoupled RoPE PPL represents the result after applying RoRoPE and FreqFold to reduce decoupled RoPE loss. Compressed KV PPL indicates the perplexity when the KV cache is compressed to 576 with the help of Balance-KV.

| Model | KV Cache | Original ppl | Decouple RoPE ppl | Compress KV ppl |
|---|---|---|---|---|
| Llama-3-8B | -71.875% | 6.1371 | 8.3997 | 18.35 |
| Qwen2.5-7B | -43.75% | 6.848 | 7.3059 | 7.9812 |
| Qwen2.5-72B | -71.875% | 4.2687 | 4.6931 | 7.7172 |
| Mistral-7B | -71.875% | 5.3178 | 5.5915 | 7.0251 |
| Mixtral-8x7B | -71.875% | 3.8422 | 4.1407 | 5.8374 |
| Gemma-2-9B-IT | -85.94% | 10.1612 | 11.421 | 21.626 |
| GPT-OSS-20B | -43.75% | 10.2563 | 9.2175 | 9.4072 |

From Table 6, we can observe that for GQA models—ranging from 7B to 70B, whether dense models or MoE models—TransMLA consistently incurs only minimal performance loss. It's worth noting that MLA is specifically optimized for a latent dimension of 576, which we adopt as a unified

compression target across all models. Since GQA models typically maintain smaller KV caches than MHA, converting them to MLA results in less information loss, making the transition both effective and efficient.

## K    Distillation Experiments

In Table 1, our work follows the experimental setup of MHA2MLA and uses the smollm-corpus for pretraining. However, this dataset differs from the original training data of LLaMA-2-7B. To investigate whether the observed improvements stem from additional learning or genuine recovery of model capabilities, we adopted the data distillation approach proposed in the LLM-QAT [46]. Specifically, we generated a new dataset from the original model to guide the distillation of the converted model. For each model, we generated distillation datasets containing 14,000 samples, with each sample consisting of 2,048 tokens. These datasets were used to distill the converted model by minimizing the error between the output of each converted MLA attention module and the corresponding GQA outputs of the original model, as well as aligning the final token predictions. Each attention module was trained for 10 epochs and hyperparameter tuning was conducted on 10 evenly spaced learning rates within the range [1e-3, 2e-2]. The results are presented in Table 7.

Table 7: Commonsense reasoning accuracy for distilling converted MLA-based models from original GQA models

| Model | KV Cache | AVG | ARC | HS | MMLU | OBQA | PIQA | WG |
|---|---|---|---|---|---|---|---|---|
| LLaMA-2-7b | — | 59.81 | 59.31 | 73.13 | 41.35 | 41.60 | 78.40 | 65.04 |
| TransMLA | -68.75% | 59.89 | 59.82 | 72.43 | 41.67 | 42.60 | 78.18 | 64.64 |
| Qwen2.5-7b | — | 63.38 | 64.45 | 77.49 | 47.26 | 45.40 | 79.76 | 65.90 |
| TransMLA | -71.875% | 63.12 | 68.13 | 74.90 | 46.33 | 44.20 | 79.11 | 66.06 |
| Mistral-7b-v0.1 | — | 64.83 | 66.71 | 79.48 | 45.67 | 45.60 | 82.86 | 68.67 |
| TransMLA | -50% | 64.48 | 66.22 | 79.00 | 45.53 | 45.60 | 82.70 | 67.80 |

According to Table 7, aligning the converted models using only the distilled data from the original models allows them to closely match the performance of the original models. This confirms that the observed performance improvements are not due to additional learning but reflect genuine restoration. We plan to distill more data and complete experiments on additional models and compression rates in the future.

## L    Ablation Study: Calibration Dataset

To investigate the impact of the calibration dataset on the effectiveness of TransMLA, we conducted an ablation study focusing on different calibration data sources and dataset sizes. As shown in Table 8, the number of calibration samples noticeably affects the conversion quality, with 64 samples achieving the best overall performance. Using different datasets for calibration also introduces slight variations in results; however, all calibration datasets outperform the baseline that uses randomly generated tokens.

## M    Ablation Study: Core Components

In Figure 3b and Figure 4b, we presented ablation results on perplexity (PPL), focusing on RoRoPE, FreqFold, and BalanceKV. In this section, we further evaluate commonsense reasoning accuracy, with the results summarized in Table 9:

- Removing RoRoPE leads to an accuracy drop of approximately 20% compared to the best-performing configuration.
- Disabling FreqFold (i.e., setting FreqFold = 1) results in about a 6.5% reduction in accuracy.
- Excluding BalanceKV causes an accuracy decline of around 10%.

These findings underscore the significance of each component in sustaining strong downstream reasoning performance.

Table 8: Ablation study of using different calibration data sources and dataset sizes for converting LLaMA-3-8B to MLA.

| Dataset | n_samples | Decouple RoPE ppl | Compress KV ppl |
|---------|-----------|-------------------|-----------------|
| wikitext2 | 1 | 8.674 | 155.65 |
| | 2 | 8.217 | 38.134 |
| | 4 | 8.192 | 22.859 |
| | 8 | 8.050 | 18.688 |
| | 16 | 8.164 | 17.499 |
| | 32 | 8.217 | 15.966 |
| | 64 | 8.368 | 14.898 |
| | 128 | 8.404 | 15.080 |
| ptb | 128 | 8.209 | 19.394 |
| c4 | 128 | 7.341 | 16.018 |
| alpaca | 128 | 7.389 | 17.398 |
| random | 128 | 15.693 | 44.956 |

Table 9: Ablation study of removing each method and its impact on commonsense reasoning performance of the converted model without training.

| Method | Avg. | MMLU | ARC | PIQA | HS | OBQA | WG |
|--------|------|------|-----|------|-----|------|-----|
| LLaMA-3-8B | 63.92 | 46.13 | 65.81 | 80.79 | 76.26 | 45.40 | 69.14 |
| TransMLA | **55.12** | **37.47** | **54.31** | **73.78** | **65.72** | **38.80** | **60.62** |
| - RoRoPE | 35.59 | 25.64 | 28.91 | 55.77 | 28.06 | 24.00 | 51.14 |
| - FreqFold | 48.63 | 31.19 | 45.24 | 69.37 | 56.18 | 34.60 | 55.17 |
| - BKV | 44.91 | 29.19 | 37.81 | 65.56 | 49.89 | 32.20 | 54.78 |

In the main paper, the default scaling factor for BalanceKV is computed as the average L2 norm of $K_{\text{RoPE}}$ divided by the average L2 norm of V. Table 10 presents an ablation study on the impact of the BalanceKV scaling factor on the performance of the converted model.

Table 10: Ablation study on the impact of the BalanceKV scaling factor on the performance of the converted model. Lower perplexity (PPL) indicates better performance.

| BalanceKV Scaling Factor | Lora PPL (WikiText2) |
|--------------------------|----------------------|
| None | 28.8341 |
| $(k_{\text{rope\_mean\_norm}} \times 4)/v_{\text{mean\_norm}}$ | 18.8790 |
| $(k_{\text{rope\_mean\_norm}} \times 2)/v_{\text{mean\_norm}}$ | 15.3617 |
| $k_{\text{rope\_mean\_norm}}/v_{\text{mean\_norm}}$ | **15.0796** |
| $k_{\text{rope\_mean\_norm}}/(v_{\text{mean\_norm}} \times 2)$ | 17.8757 |
| $k_{\text{rope\_mean\_norm}}/(v_{\text{mean\_norm}} \times 4)$ | 27.9021 |
| Optimized per-layer | **13.7945** |

On LLaMA-3-8B, completely removing BalanceKV results in a 3.8-point increase in perplexity, highlighting its critical role in maintaining model quality. We also observed that model performance is sensitive to the specific scaling coefficient used in BalanceKV. The default value reported in our paper already yields strong overall performance. To explore this further, we performed a per-layer grid search over the range [0.3, 2.0] with 20 intervals to minimize the output difference from the original model. This tuning produced a 1.2-point reduction in perplexity compared to the default configuration, suggesting that layer-wise coefficient optimization is both feasible and beneficial for further performance improvement.

# N   Evaluation on the LongBench Benchmark

TransMLA achieves KV cache compression by reducing the dimensionality of token representations, making it orthogonal to methods such as token pruning and quantization. However, as sequence length increases, the compression error for each token can accumulate. To evaluate the effectiveness

of TransMLA on long-context tasks, we conducted experiments using Qwen-2.5-7B, which supports a maximum positional embedding length of 131,072. The model was converted into an MLA-based architecture, and the original model was then used to distill the converted one. We report the performance of both the original and converted (distilled) models across multiple long-context tasks from the LongBench benchmark in Table 11.

Table 11: Performance on multiple long-text tasks and inference speed (tested on 16k-length context) of Qwen after converting GQA to MLA and applying further distillation. Abbreviations: **SD-QA** = Single-Document Question Answering, **MD-QA** = Multi-Document Question Answering, **Sum.** = Summarization, **FSL** = Few-Shot Learning, **Code** = Code Completion, **Avg.** = Average across all tasks.

| Model | Thpt. | SD-QA | MD-QA | Sum. | FSL | Code | Avg. |
|---|---|---|---|---|---|---|---|
| Qwen2.5-7B | 1233.04 | 20.29 | 25.73 | 24.46 | 62.11 | 63.10 | 39.14 |
| TransMLA | 1925.19 | 16.50 | 19.02 | 18.51 | 44.20 | 38.39 | 27.32 |
| TransMLA-distill | 1925.19 | 18.74 | 28.26 | 22.39 | 61.00 | 59.21 | 37.92 |

While TransMLA provides substantial speedup (approximately 1.56× throughput improvement), its zero-shot performance declines significantly when no additional training is applied. Although distillation can partially recover accuracy, performance restoration becomes increasingly difficult on long and complex reasoning tasks. Future work will focus on reducing the compression-induced degradation in TransMLA, aiming to achieve stronger zero-shot generalization on challenging long-context benchmarks.

# O    Extension to Multi-Group MLA

FlashMLA supports only a fixed configuration of 64 RoPE dimensions and 512 non-RoPE dimensions for its KV cache, which limits compression flexibility. To address this, we divide the query heads into multiple groups, converting each group into an independent MLA module.

We experimentally verified this in Table 12: under identical KV cache conditions, the ungrouped approach outperforms the two-group version. However, the grouped method enables the model to retain more KV cache, so it performs better than the original TransMLA with a single 576-dimensional KV cache.

Importantly, this multi-group strategy remains compatible with FlashMLA inference. We tested the inference speeds of MHA, GQA, two-group MLA, and single-group MLA under a 16k context length. Since FlashMLA does not support inference with KV cache dimensions other than 576, we excluded the single-group 64+1024 MLA configuration. As shown in Table 12, the two-group MLA achieves a 6.2× speedup over MHA and a 1.58× speedup over GQA, while the single-group MLA remains approximately 1.9× faster than the two-group version.

Table 12: Performance comparison of multi-group MLA under different RoPE/NoPE dimensions and group settings. Abbreviations: **rope_dim** = RoPE dimension, **nope_dim** = NoPE dimension, **num_grp** = number of MLA groups, **ppl** = perplexity, **Thpt.** = throughput in tokens per second (tested at 16k context length).

| Model | rope_dim | nope_dim | num_grp | ppl | Thpt. (tok/s) |
|---|---|---|---|---|---|
| LLaMA-2-7B | - | - | - | 5.4732 | 11776 |
|  | 64 | 512 | 1 | 41.6135 | 141246 |
|  | 64 | 1024 | 1 | 24.0931 | – |
|  | 64 | 512 | 2 | 31.2177 | 73098 |
| LLaMA-3-8B | - | - | - | 6.1371 | 46247 |
|  | 64 | 512 | 1 | 25.8047 | 141246 |
|  | 64 | 1024 | 1 | 8.9768 | – |
|  | 64 | 512 | 2 | 12.9472 | 73098 |

# P Case Study

To provide an intuitive understanding of TransMLA's impact on model performance, this section presents several examples from vLLM's docs. We compare the outputs of three model variants: (1) a model with 92.97% of its KV cache compressed without any fine-tuning; (2) a model pretrained on 6B tokens, as detailed in Table 1; and (3) a model fine-tuned for one epoch on the SmolTalk dataset, following the setup described in [47]. The results are summarized in Table 13.

As shown in Table 13, even without any additional training, the compressed model is still able to produce coherent and meaningful responses. This demonstrates the effectiveness of techniques such as RoRoPE, FreqFold, and BKV-PCA in significantly mitigating performance degradation. Moreover, with a modest amount of pretraining or supervised fine-tuning (SFT), the model's performance improves substantially. These findings highlight TransMLA's potential as a general framework for converting various GQA models into MLA models, with promising prospects for aligning with the performance of advanced systems like DeepSeek R1.

Table 13: Examples from different model configurations. Red indicates input; black indicates output. "w/o Training" denotes the TransMLA-compressed model (92.97% KV cache) without further training. "Pre-Training" and "Fine-Tuning" show outputs after pretraining on a 6B-token corpus and SFT on SmolTalk [48], respectively.

| Model | Prompt & Generated Text |
|---|---|
| w/o Training | Hello, my name is Katiu, my father's dog, the pet of the 3600 year-old tribe, Kint. The Kangs were part of a race of reptiles. A small handful |
| Pre-Training | Hello, my name is Sasha and I am in third grade at Meadows. You may be wondering what this article is about. Well, I have been doing a lot of research on the water cycle and decided to write about it. |
| Fine-Tuning | Hello, my name is Emily, and I'm a 20-year-old college student. My hobbies include painting, writing, and photography. I also enjoy playing the guitar. |
| w/o Training | The president of the United States is elected by the legislature. The legislature controls the national armed forces, but only provides the funds to establishing a national guard. |
| Pre-Training | The president of the United States is elected to a four-year term by the people of each state in a general election held every four years on the Tuesday following the first Monday in November. |
| Fine-Tuning | The president of the United States is not a position to be taken lightly. This person is the chief executive of the United States of America, and has immense power and influence. |
| w/o Training | The capital of France is Paris. Its geographical position in the Iberian Plain of France, Spain, Spain, and Morocco are the four largest cities. This region is located in Asia, Spain and Morocco. |
| Pre-Training | The capital of France is a major business city and it is a favorite destination for businesses from all over the world. It has a strategic location in the heart of the European Union, which makes it one of the most popular cities in Europe. |
| Fine-Tuning | The capital of France is Paris, and it is one of the most popular tourist destinations in the world. It is a city that offers something for everyone, from art and history to food and fashion. |
| w/o Training | The future of AI is in serious risk to create a major breakthrough in this emerging phenomenon in the history of artificial intelligence. |
| Pre-Training | The future of AI is looking bright. With advancements in technology and the increasing availability of data, AI is expected to become more intelligent and capable of performing even more complex tasks. |
| Fine-Tuning | The future of AI is The future of AI is more nuanced and complex than we might think. Here are some potential developments that could shape the future of AI. |

