# OpenReview forum: "TransMLA: Migrating GQA Models to MLA with Full DeepSeek Compatibility and Speedup"
_NeurIPS.cc/2025/Conference — NeurIPS 2025 spotlight_

### Official Review · Reviewer_E7fv · 2025-06-05

**Clarity:** 3
**Significance:** 4
**Originality:** 3
**Rating:** 5
**Confidence:** 4

**Summary:**

This paper presents TransMLA, a framework for converting GQA-based language models (e.g., LLaMA, Qwen) into the MLA (Multi-head Latent Attention) format used by DeepSeek models. TransMLA enables off-the-shelf migration of pre-trained GQA models into the MLA structure, thereby inheriting the benefits of DeepSeek’s inference-time acceleration (e.g., vLLM, SGlang, FP8). The authors propose several innovations to support this conversion: (1) RoRoPE, a PCA-based method to concentrate RoPE positional information into a small number of dimensions, (2) FreqFold, which exploits frequency similarity in RoPE to further reduce redundancy, and (3) Balanced Key-Value PCA, which equalizes key and value norms to enable better joint compression. Extensive experiments on smolLM and LLaMA-2 demonstrate that TransMLA achieves substantial compression (up to 93%) and speedup (up to 10.6×) while preserving model quality with minimal fine-tuning.

**Questions:**

Can the authors release the converted MLA models (e.g., LLaMA2 7B)? This would greatly benefit the community and validate the plug-and-play nature of TransMLA.

How sensitive is Balanced KV to the choice of scaling constant? Could this be learned or optimized per-layer?

**Ethical Concerns:**

["NO or VERY MINOR ethics concerns only"]

**Final Justification:**

The rebuttal it supplies a theoretical proof for Balanced KV, offers privacy-safe alternatives of distilled data or weight-only SVD, and clarifies RoRoPE/FreqFold with new visual aids and shows a per-layer scaling search that improves perplexity to confirm Balanced KV’s importance. My concerns has been mostly resolved and I maintain my score of 5.

**Limitations:**

Yes

**Paper Formatting Concerns:**

No major formatting issues

**Quality:**

4

**Strengths And Weaknesses:**

**Strengths**

Strong practical relevance: This work addresses a highly relevant engineering challenge to migrate existing GQA models to DeepSeek’s efficient MLA format without retraining from scratch.

Well-motivated and well-engineered: The techniques such as RoRoPE and Balanced KV are practical, intuitive, and empirically effective.

Thorough empirical evaluation: The paper includes both performance recovery metrics and inference latency benchmarks across multiple hardware setups, providing a well-rounded view of the benefits.

Substantial performance gain: At similar compression rates, TransMLA clearly outperforms MHA2MLA both in accuracy and speed, especially under high compression ratios.

Compatibility with ecosystem: Seamless integration with DeepSeek’s toolchain is a strong point, making this work immediately usable for real-world deployment.

**Weaknesses**

Balanced Key-Value PCA is heuristic: While intuitive and empirically validated, the balancing method lack theoretical guarantees or principled formulation (e.g., via optimization objectives).

Compression calibration requires data: The PCA-based compression still relies on a calibration dataset for collecting activations, which may not be trivial in privacy-sensitive or black-box model settings.

Clarity in exposition: Some parts, such as RoPE slicing and rotation math (Section 4.2), are quite dense and could benefit from visual aids or step-by-step walkthroughs.

---

> ### Author Rebuttal · Authors · 2025-07-31
>
> Thank you very much for your encouraging review and constructive feedback. Below we provide point-by-point responses to your comments. All suggested improvements will be incorporated into the camera-ready version.
>
> **Q1. While intuitive and empirically validated, the balancing method lacks theoretical guarantees or principled formulation.**
>
> **A1.** Balanced Key-Value PCA (BKV) is indeed a simple yet effective technique. While many prior works (Palu, MHA2MLA, X-EcoMLA) have not adopted this method, we highlight it as a novel contribution. Behind its apparent simplicity, BKV has a **solid theoretical foundation**. We prove that under a joint loss function minimizing the relative approximation errors of both $W_K$ and $W_V$ components:
> $\mathcal{L} = \|| \frac{W_K - W_K^{top-r}}{\mathrm{RMS}(W_K)} \||^2 +\|| \frac{W_V - W_V^{top-r}}{\mathrm{RMS}(W_V)} \||^2$, scaling $W_K$ and $W_V$ to the same magnitude: $W_{\text{scaled}}^{KV} =\(W_K / \mathrm{RMS}(W_K), W_V / \mathrm{RMS}(W_V)\)$ achieves the optimal low-rank approximation under this specific loss function. In our proof, we use $a$ and $b$ as scaling factors. We first show that the loss function depends only on the ratio $a:b$, not on the absolute values of $a$ and $b$. By setting $b=1$, the optimization simplifies to finding the optimal value for $a$. Next, we compute the derivative of the loss function with respect to $a$, and show that: $\frac{d\mathcal{L}(a, 1)}{da} = 0$ when $a = \mathrm{RMS}(W_K) / \mathrm{RMS}(W_V)$, through a series of substitutions. The main substitutions used are:
> $\||W\||^2 = \mathrm{tr}(W^T W) = \sum_{i=1}^{\mathrm{rank}(W)} \sigma_i^2, \quad \text{and} \quad \frac{d\sigma_i}{da} = u_i^T \frac{dW}{da} v_i$,
> where $u_i$, $\sigma_i$, and $v_i$ are the left singular vectors, singular values, and right singular vectors of matrix $W$, respectively.
> This proof provides a theoretical foundation and a more principled mathematical formulation for our Key-Value balancing strategy. The detailed proof and mathematical derivation will put in the appendix of the camera-ready version.
>
> **Q2. On the need for a calibration dataset and concerns in privacy-sensitive or black-box settings:**
>
> **A2:** Regarding privacy-sensitive datasets, please see our response to **Reviewer qHce A2**: we address this by using distilled data generated from the original model, which avoids direct access to private corpora.
>
> As for black-box models, TransMLA is primarily intended to accelerate inference, which generally requires access to model internals. However, we will fully open-source TransMLA, including conversion, training, and inference code, as well as converted checkpoints. This allows model providers to apply the method internally without data sharing. Alternatively, they may use our converted foundation models and fine-tune them on in-house data.
>
> In addition, we also provide alternative implementations of RoRoPE, FreqFold, and BKV that do not require calibration data. Instead of performing PCA on activations, these versions apply SVD directly to the weight matrices. This option will be included in our open-sourced codebase for users who prefer not to rely on calibration datasets.
>
> **Q3. On clarity in exposition, especially for RoPE slicing and rotation math in Section 4.2:**
>
> **A3.** Thank you for this valuable suggestion. Although we made a conscious effort to ensure readability during writing, we recognize that several theoretical components in TransMLA—especially **RoRoPE**—remain abstract and challenging to follow. We have since redesigned the **conversion pipeline diagram** to include each component’s purpose. Additionally, we created intuitive **frequency-domain visualizations** to explain both RoRoPE and FreqFold. These updates significantly improve clarity and allow readers to grasp the motivation even without diving into all the formulas. These diagrams will be included in the camera-ready version.
>
> **Q4. On releasing converted MLA models such as LLaMA2-7B:**
>
> **A4.** Absolutely. We will release all code for conversion, training, and inference, as well as checkpoints for converted models. In the future, we plan to support more models and more tasks, providing the community with faster and stronger **pretrained models.**
>
> **Q5. On the sensitivity of Balanced KV to the scaling constant and potential for optimization:**
>
> **A5.** To investigate this, we conducted an ablation study as shown in Table 8. On LLaMA-3-8B, completely removing Balanced KV results in a 3.8-point increase in perplexity, highlighting its critical role. We also observed that model performance is sensitive to the scaling coefficient used in Balanced KV. The default value in our paper already yields strong overall results. However, following your suggestion, we performed a per-layer grid search over the range [0.3, 2.0] with 20 intervals, aiming to minimize the output difference from the original model. This tuning led to a 1.2-point reduction in perplexity compared to the default, suggesting that layer-wise coefficient optimization is both feasible and beneficial for further improving performance.
>
> **Table 8.** Ablation study on the impact of the Balance KV scaling factor on the performance of the converted model. In the main paper, the default factor is computed as the average L2 norm of $K_{\text{RoPE}}$ divided by the average L2 norm of V.
>
> | BKV | lora ppl (wiki) |
> | --- | --- |
> | None | 28.8341 |
> | (k_rope_mean_norm*4)/v_mean_norm | 18.8790 |
> | (k_rope_mean_norm*2)/v_mean_norm | 15.3617 |
> | k_rope_mean_norm/v_mean_norm | **15.0796** |
> | k_rope_mean_norm/(v_mean_norm*2) | 17.8757 |
> | k_rope_mean_norm/(v_mean_norm*4) | 27.9021 |
> | optimized per-layer | **13.7945** |

---

### Official Review · Reviewer_KGKd · 2025-07-02

**Clarity:** 2
**Significance:** 3
**Originality:** 3
**Rating:** 4
**Confidence:** 4

**Summary:**

In this paper, the authors propose TransMLA, a framework that converts GQA/MHA modules from existing LLM models (such as Llama) into MLA modules that fit DeepSeek's frameworks. To adopt the RoPE from GQA/MHA to the decoupled RoPE in DeepSeek's MLA, the authors propose RoRoPE and FreqFold to concentrate the RoPE information to the first key/query head. Besides, a balanced key-value method is proposed for model compression.  The proposed method is shown to be a better conversion than existing works (such as MHA2MLA) and could greatly save the memory cost of the KV cache.

**Questions:**

1. Overall I think Figure 1 is relatively clear to me but they are not well-explained in Section 4.
 - For example, it is not known how $q_{rope}$, $q_{nope}$, $k_{rope}$, $k_{nope}$ are defined in TransMLA.
 - It is not known how the RoPE is removed for $q_{nope}$ and  $k_{nope}$ after applying the PCA rotation matrix.
 - Also, it is not know how the matrices in MLA are exactly initialized (e.g., $W_{dq}$, $W_{qr}$, $W_{uq}$, $W_{dkv}$, $W_{uk}$, $W_{uv}$)

2. The sum sign seems to be missing in Equation (29)

3. It is a little bit not clear to me how this claim "MLA consistently exhibits greater expressive capacity than GQA given the same KV cache size" helps in this paper. When setting the same KV cache size, MLA seems clearly stronger than GQA as there are more parameters. It would be great to know if there are any study/theoriem comparing the expressive capacity of GQA and MLA given similar amount of parameters.

4. There seems to be another related work named X-EcoMLA[1]. How is the proposed TransMLA compared to this work?

5. I am wondering how the authors benefit from the DeepSeek optimizations for the inference speed measurement. Is the FlashMLA adopted in this experiment? It would be beneficial to add more details for the inference optimization.

[1] Li, Guihong, et al. "X-ecomla: Upcycling pre-trained attention into mla for efficient and extreme kv compression." arXiv preprint arXiv:2503.11132 (2025).

**Ethical Concerns:**

["NO or VERY MINOR ethics concerns only"]

**Final Justification:**

I appreciate the author's responses, which have solved most of my concerns. I have raised my score.

**Limitations:**

Yes

**Quality:**

2

**Strengths And Weaknesses:**

Strengths:
- The proposed method achieves much better performance than MHA2MLA, especially when there is no additional training. This means the proposed TransMLA is a better approximation to existing MHA/GQA and can be used as a starting point for future research

Weakness:
- Although the proposed TransMLA achieves good performance, it requires a calibration dataset for PCA and the effect of the calibration dataset is not studied. For example, what is the effect of the size of the calibration dataset? Also, what if we choose another dataset as the calibration dataset?
- The description of the proposed method is not very clear, and there are some typos (please refer to Questions)
- There is a lack of ablation studies. For example, what would be the commonsense reasoning performance if we don't enable all the components (e.g., RoRoPE without FreqFold)
- The authors missed some other related works in the literature (please refer to Questions)
- The authors tested TransMLA on SmolLM-1.7B and Llama2-7B, but there is no experiment on Qwen/Mistral (which is claimed in the abstract)
- The proposed method is not compared to other KV cache compression methods such as KVquant and Palu
- The proposed method is only tested on common-sense reasoning tasks. It would be good to test on some long-context tasks (such as LongBench) since MLA's advantage is mainly on long context lengths

---

> ### Author Rebuttal · Authors · 2025-07-31
>
> We sincerely thank you for your detailed questions and constructive feedback. Below, we address each of your concerns and will incorporate the necessary updates in the camera-ready version.
>
> **Q1. Calibration dataset for PCA: effect of size and dataset choice**
>
> **A1.** As per your suggestion, we conducted an ablation study on the calibration dataset. As shown in **Table 4**, the number of calibration samples does affect the conversion quality, with **64 samples yielding the best performance**. Using different datasets for calibration also impacts results slightly, but all perform better than using randomly generated tokens.
>
> **Table 4.** Ablation study of using different calibration data sources and dataset sizes for converting LLaMA-3-8B to MLA.
>
> |  | n_samples | Decouple RoPE ppl | Compress KV  ppl |
> | --- | --- | --- | --- |
> | wikitext2 | 1 | 8.674 | 155.65 |
> |  | 2 | 8.217 | 38.134 |
> |  | 4 | 8.192 | 22.859 |
> |  | 8 | 8.05 | 18.688 |
> |  | 16 | 8.164 | 17.499 |
> |  | 32 | 8.217 | 15.966 |
> |  | 64 | 8.368 | 14.898 |
> |  | 128 | 8.404 | 15.08 |
> | ptb | 128 | 8.209 | 19.394 |
> | c4 | 128 | 7.341 | 16.018 |
> | alpaca | 128 | 7.389 | 17.398 |
> | random | 128 | 15.693 | 44.956 |
>
> **Q2. Lack of ablation studies (e.g., commonsense** reasoning performance for **RoRoPE without FreqFold)**
>
> A2. We have presented ablations on perplexity (PPL) in **Figure 2 and Figure 3** of our submission for RoRoPE, FreqFold, and BalanceKV. Following your suggestion, we additionally evaluated **commonsense reasoning accuracy**, and the results are shown in **Table 5**:
>
> - Without **RoRoPE**, accuracy drops by ~20% compared to the best configuration.
> - Without **FreqFold** (e.g. FreqFold = 1), accuracy drops by ~6.5%.
> - Without **BalanceKV**, accuracy drops by ~10%.
>
> These results highlight the importance of each component in maintaining strong downstream performance.
>
> **Table 5.** Ablation study of removing each method and its impact on commonsense reasoning performance of the converted model without training.
>
> |  | Avg. | MMLU | ARC | PIQA | HS | OBQA | WG |
> | - | - | --- | --- | --- | --- | --- | --- |
> | llama-3-8B | 63.92 | 46.13 | 65.81 | 80.79 | 76.26 | 45.40 | 69.14 |
> | TransMLA | **55.12** | **37.47** | **54.31** | **73.78** | **65.72** | **38.80** | **60.62** |
> | - RoRoPE | 35.59 | 25.64 | 28.91 | 55.77 | 28.06 | 24.00 | 51.14 |
> | - FreqFold | 48.63 | 31.19 | 45.24 | 69.37 | 56.18 | 34.60 | 55.17 |
> | - BKV | 44.91 | 29.19 | 37.81 | 65.56 | 49.89 | 32.20 | 54.78 |
>
> **Q3. Experiments on Qwen / Mistral**
>
> **A3.** As mentioned in  **Reviewer qHce A1**, our work follows the experimental setup of MHA2MLA and uses the smollm-corpus for pretraining SmolLM-1.7B and LLaMA-2-7B. To answer your question, we converted Qwen and Mixtral to MLA and conducted distillation experiments to recover the performance. The results are shown in the following table.
>
> | Model | KV cache | avg | arc | hellaswag | mmlu | openbook_qa | piqa | winogrande |
> | - | - | - | - | - | - | - | - | - |
> | LLaMA-2-7b | — | 0.5981 | 0.5931 | 0.7313 | 0.4135 | 0.4160 | 0.7840 | 0.6504 |
> | TransMLA |  -68.75% | 0.5989 | 0.5982 | 0.7243 | 0.4167 | 0.4260 | 0.7818 | 0.6464 |
> | Qwen2.5-7b | — | 0.6338 | 0.6445 | 0.7749 | 0.4726 | 0.4540 | 0.7976 | 0.6590 |
> | TransMLA | -71.875% | 0.6312 | 0.6813 | 0.7490 | 0.4633 | 0.4420 | 0.7911 | 0.6606 |
> | Mistral-7b-v0.1 | — | 0.6483 | 0.6671 | 0.7948 | 0.4567 | 0.4560 | 0.8286 | 0.6867 |
> | TransMLA | -50% | 0.6448 | 0.6622 | 0.7900 | 0.4553 | 0.4560 | 0.8270 | 0.6780 |
>
>
> **Q4. Comparison with KVQuant and Palu**
>
> **A4.** KVQuant is discussed in **Line 34** of our original paper, and Palu is discussed in **Lines 109–114**. As further explained in **Reviewer i2PF’s comments A1 (second paragraph) and A3.2**, TransMLA is orthogonal to KVQuant and can be combined with it. Palu avoids decoupling RoPE by compressing only modules that do not include positional encoding. In their Figure 5, they show that without RoPE, *absorb* enables a **6.17×** speedup comparing to MHA module at a 64K context length. However, when RoPE is present, *absorb* is disabled, and the speedup drops to **2.91×**. This highlights the significance of our contribution, which enables low-rank compression of RoPE—making *absorb* usable and thus enabling more efficient inference.
>
> **Q5. Test on LongBench benchmark**
>
> Following your suggestion, we select Qwen-2.5-7B (with max position embeddings of 131072) and convert it into an MLA model. The original model is then used to distill the converted one. We report the performance of both the original and the converted (distilled) models on multiple long-context tasks from the **LongBench** in Table 6.
>
> **Table 6.** Performance on multiple long-text tasks and speed (tested on 16k-length context) of Qwen after converting GQA to MLA and applying further distillation, as described in **Reviewer qHce A2**.
>
> |  | Throughput(tokens/s) | Single-Document QA | Multi-Document QA | Summarization | Few-Shot Learning | Code Completion | Avg. |
> | - | - | - | - | - | - | - | - |
> | Qwen2.5-7B | 1233.04 | 20.29 | 25.73 | 24.46 | 62.11 | 63.10 | 39.14 |
> | TransMLA | 1925.19 | 16.50 | 19.02 | 18.51 | 44.20 | 38.39 | 27.32 |
> | TransMLA-distill | 1925.19 | 18.74 | 28.26 | 22.39 | 61.00 | 59.21 | 37.92 |
>
> **Q6. Figure 1 is not well-explained.**
>
> **A6.** Thank you for pointing this out. We realized that Figure 1 contained too much information, so we have split it into three separate parts: an overall workflow diagram, a frequency-domain transformation illustration for RoRoPE, and a schematic of the FreqFold approximation. Each annotated component in the figures is now explained step-by-step in the Method section. These updates will be included in the camera-ready version.
>
> **Q7. How does proving that MLA has stronger expressive capacity than GQA given the same KV cache size or parameters help support the contributions of this paper?**
>
> **A7:** Please refer to **Reviewer i2PF A1.1 (first paragraph)** and **A3.1**. “MLA consistently exhibits greater expressive capacity than GQA given the same KV cache size” is a **central motivation** of this conversion.
>
> **Q8. MLA seems clearly stronger than GQA as there are more parameters.**
>
> **A8.** More parameters do not necessarily imply stronger expressive capacity across different model architectures. For example, as noted in the original DeepSeek paper, MHA with more parameters demonstrates lower expressive capacity than MLA. One of the key contributions of our paper is a constructive proof showing that, under equal KV cache size, all GQA can be converted into MLA, but the reverse is not true. This provides a formal guarantee that MLA always possesses stronger expressive capacity than GQA.
>
> **Q9. Any other study comparing the expressive capacity of GQA and MLA.**
>
> **A9.** Since the introduction of MLA last year, several studies have proposed hypotheses and conducted experiments to explore the source of MLA’s strong performance. For example, **Jianlin Su** (the author of RoPE) discussed on his blog that increasing the **head dimensions** in MLA may contribute to its improved expressiveness.
>
> **Q10. Comparison with X-EcoMLA**
>
> **A10.** X-EcoMLA removes all positional encodings in the low-rank (nope) part and averages across all heads to extract 64-dimensional RoPE embeddings. This approach lacks theoretical justification for its approximation validity. Both the rope and nope parts of their transformation introduce significant performance drop. As X-EcoMLA is not open-sourced yet, we were unable to reproduce and benchmark their method. Once their code is available, we will provide a direct comparison in the camera-ready version.
>
> **Q11.** I am wondering how the authors benefit from the DeepSeek optimizations for the inference speed measurement. Is the FlashMLA adopted in this experiment? It would be beneficial to add more details for the inference optimization.
>
> **A11.** FlashMLA supports only the Hopper architecture. In our original submission, we did not have access to Hopper GPUs, so we used VLLM with Triton backend for MLA, and FlashAttention backend for MHA. During the rebuttal period, we reran the experiments using Hopper GPUs with FlashMLA backend. As shown in Table 7, compared to the original GQA model on the same GPU, FlashMLA achieves up to an **11.32× speedup at 32K context length**. The detailed information and the code will be publicly available on the camera ready paper.
>
> **Table 7.** Throughput (tokens/s) comparison between LLaMA-2-7B and the converted MLA-based model, evaluated on Hopper architecture.
>
> |  | 1K | 2K | 4K | 8K | 16K | 32K |
> | - | - | - | - | - | - | - |
> | LLaMA-2-7b | 3943.49 | 2122.85 | 1252.37 | 639.32 | 323.58 | 158.37 |
> | TransMLA | 10967.49 | 6461.55 | 5702.00 | 4679.00 | 3436.58 | 1792.79 |
> | Speedup | 2.78x | 3.04x | 4.55x | 7.32x | 10.62x | 11.32x |

---

> > ### Author Response · Authors · 2025-08-05
> > **We look forward to your response.**
> >
> > Dear Reviewer KGKd,
> >
> > Thank you for your thorough reading of our article and for providing many helpful suggestions. We also appreciate your recognition that TransMLA is a better approximation of the existing MHA/GQA and could serve as a starting point for future research.
> >
> > In response to the seven weaknesses you highlighted, we have conducted additional comparative experiments, which we believe will enhance the credibility of TransMLA’s performance. These ablation experiments will be included in the camera-ready version of the paper. We appreciate your valuable feedback.
> >
> > Furthermore, we have addressed the five questions you raised and will provide more detailed explanations of the methods and experiments in the camera-ready version. We hope these revisions will further improve the clarity and reproducibility of our work.
> >
> > We trust that our responses have adequately addressed your comments and questions, and we are happy to answer any additional ones.
> >
> > If you find our clarifications satisfactory, we would greatly appreciate your consideration of raising your rating of our work.
> >
> > Best regards,
> >
> > Submission 3063 Authors

---

> > ### Comment · Reviewer_KGKd · 2025-08-06
> > **Thanks for the rebutal**
> >
> > I want to thank the authors for their detailed responses and new results. Many of my concerns have been resolved. I have some remaining questions:
> > - It is great to see that the proposed method could apply to Qwen. I am wondering how TransMLA approximates the qkv bias term in Qwen2.5-7B. Will that require significant changes to the algorithm?
> > - I appreciate that the authors did additional experiments with FlashMLA and the results look good. One question I have is that FlashMLA seems to only support the DeepSeekV3 dimensions. When the number of heads and kv_rank of TransMLA are different from DeepSeekV3, how to support TransMLA with FlashMLA?
> > - This is a minor point. Regarding A8, I agree that more parameters do not translate to more expressive power but it also depends on how many more parameters. When comparing GQA and MLA with the same KV cache budget, MLA seems to have more than 10% parameters which is not negligible. I agree with the motivation of TransMLA, and I believe it is a useful technique. However, I think it is unfair and not very useful to analyze and compare the express power of GQA and MLA with the same KV cache size. When applying TransMLA to Llama or Qwen, the KV cache of TransMLA will always be smaller.

---

> ### Author Response · Authors · 2025-08-06
> **Thank you for your response**
>
> We first address your new questions one by one:
>
> **Q1. It is great to see that the proposed method could apply to Qwen. I am wondering how TransMLA approximates the qkv bias term in Qwen2.5-7B. Will that require significant changes to the algorithm?**
>
>  **A1.** Thank you for your insightful question. TransMLA can be seamlessly applied to Qwen-2-7B. Regarding the bias in the qkv projection, we employ a simple yet effective approach: for any linear module with bias $Y = WX + b$, where $X \in \mathbf{R}^d$ is the input activation, we transform it into a bias-free form by defining $X' = (X, 1) \in \mathbf{R}^{d+1}$ and $W' = (W, b)$. This allows us to express the operation as $Y = W'X'$ without bias. Consequently, we can directly apply RoRoPE, FreqFold, and BKV techniques without additional complexity. We will provide more detailed explanations of this approach in the camera-ready version of our paper.
>
> **Q2. I appreciate that the authors did additional experiments with FlashMLA, and the results look good. One question I have is that FlashMLA seems to only support the DeepSeekV3 dimensions. When the number of heads and kv_rank of TransMLA are different from DeepSeekV3, how to support TransMLA with FlashMLA?**
>
>  **A2.** Thank you for raising this important question. While FlashMLA was initially designed with DeepSeekV3's 128 heads in mind, it remains fully compatible with other head configurations— Kimi-K2 (64 heads) and Llama-2-7B-TransMLA (32 heads).
> Regarding kv_rank support, the current implementation does specialize for kv_rank=576. However, to enable flexibility while maintaining acceleration benefits, we've developed a grouped MLA approach: by partitioning the kv_cache into multiple groups (each with dimension 576), we can effectively support larger total kv_rank values. For example, a kv_rank of 1088 can be achieved through two 512-dimensional groups plus a 64-dimensional remainder carrying positional information. We will release the implementation details. This design ensures that TransMLA maintains both performance and adaptability across different model architectures.
>
> **Q3. This is a minor point. Regarding A8, I agree that more parameters do not translate to more expressive power, but it also depends on how many more parameters. When comparing GQA and MLA with the same KV cache budget, MLA seems to have more than 10% parameters, which is not negligible. I agree with the motivation of TransMLA, and I believe it is a useful technique. However, I think it is unfair and not very useful to analyze and compare the express power of GQA and MLA with the same KV cache size. When applying TransMLA to Llama or Qwen, the KV cache of TransMLA will always be smaller.**
>
>  **A3.** Indeed, under the same KV cache budget, MLA does have more parameters than GQA, and in our proof, these additional parameters are a key reason for its improved capability. Since we aim to align inference speed when comparing expressive power, please refer to Reviewer i2PF's Table 3, where the relationship between KV cache and inference speed is more direct compared to the impact of parameter count.
>
> Your new questions have been extremely helpful in improving our paper, and we believe they will make TransMLA more rigorous, readable, and reproducible. Have our responses fully addressed all your concerns? Based on the original text, the additional experiments and explanations provided during rebuttal, we would be very grateful if you consider raising your score.

---

> > ### Author Response · Authors · 2025-08-07
> > **Do you find our new response satisfactory?**
> >
> > Dear Reviewer KGKd,
> >
> > Thank you for raising the questions, which helped us identify some previously overlooked issues. Your feedback has been greatly beneficial to our paper. Our responses have addressed your initial concerns, and you have further raised new questions, to which we have also provided answers. May we kindly ask if you are satisfied with the answers to the new questions?
> > We believe in TransMLA's solid theoretical innovations and practical methodological contributions. Considering that the author-reviewer discussion period is coming to an end, we kindly remind you that if you find the contributions in our paper worthy of acceptance, we would deeply appreciate your recognition.
> >
> > Best regards,
> >
> > Authors

---

> > > ### Comment · Reviewer_KGKd · 2025-08-08
> > > **Thanks for the response**
> > >
> > > I would like to thank the authors for their further responses which have resolved my concerns. I am still concerned about the FlashMLA integration part, as it is not clear whether the grouped MLA approach is equivalent to the original MLA during inference. However, in general, the rebuttal is satisfactory, and I will seriously consider raising my score.

---

> ### Author Response · Authors · 2025-08-08
> **Grouped TransMLA**
>
> Dear Reviewer KGKd,
>
> Thank you for your professional question. Indeed, grouped TransMLA is not equivalent to the original TransMLA, as performing PCA in groups yields weaker results compared to joint PCA when retaining the same feature dimensions. We have experimentally verified this in the attached table: under the same KV cache conditions, the ungrouped approach outperforms the two-group version. However, the grouped method allows for retaining more KV cache, which is why it performs better than the original TransMLA with only a single 576-dimensional KV cache.
> Importantly, this grouped approach maintains compatibility with FlashMLA inference. We tested the speeds of MHA, GQA, two-group MLA, and single-group MLA (FlashMLA does not support inference with dimensions other than 576, so we did not compare the single-group 64+1024 MLA) under 16k context length. As shown in the table, the two-group MLA achieves a 6.2× speedup over MHA and a 1.58× speedup over GQA, while the single-group MLA is 1.9× faster than the two-group version.
>
> | model | rope_dim  | nope_dim  | num_group  | ppl | throughout (tokens/s) |
> | --- | --- | --- | --- | --- | --- |
> | LLaMA-2-7b | - | - | - | 5.4732 | 11776.75  |
> |  | 64 | 512 | 1 | 41.6135 | 141246.74  |
> |  | 64 | 1024 | 1 | 24.0931 | - |
> |  | 64 | 512 | 2 | 31.2177 | 73098.44 |
> | LLaMA-3-8b | - | - | - | 6.1371 | 46247.48 |
> |  | 64 | 512 | 1 | 25.8047 | 141246.74 |
> |  | 64 | 1024 | 1 | 8.9768 | - |
> |  | 64 | 512 | 2 | 12.9472 | 73098.44 |
>
> We sincerely appreciate your satisfaction with our rebuttal and your consideration of raising the score. Given that the author-reviewer discussion will conclude in 12 hours, we would be immensely grateful if you could adjust your rating.
>
> Respectfully,
>
> Submission 3063 Authors

---

### Official Review · Reviewer_i2PF · 2025-07-03

**Clarity:** 4
**Significance:** 3
**Originality:** 3
**Rating:** 4
**Confidence:** 4

**Summary:**

The TransMLA framework aims to seamlessly convert existing GQA (Group-Query Attention) pre-trained models into MLA (Multi-head Latent Attention) models. TransMLA uses RoRoPE (Rotary Position Embedding Rotation) and FreqFold: to concentrate the rotational position encoding (RoPE) information into a small number of dimensions (especially the first attention head), thereby removing RoPE from the remaining dimensions to support DeepSeek's unique Absorb operation. It is observed that the norm of the key after removing RoPE is significantly larger than the value. The authors balance the norms of the two by scaling to improve the compression performance of the joint low-rank decomposition. TransMLA achieves significant KV cache compression and inference acceleration.

**Questions:**

Is MLA really better than GQA? The paper only proves the phenomenon of equal KV Cache, but does it have an advantage under equal parameter values ​​or headdim? Or is GQA good enough for us? There are many ways to solve KVCache, not necessarily MLA.

If MLA is indeed theoretically superior, and DeepSeek actively open-sources its optimization solution, then why does the GQA model still dominate and require such a complex conversion?

**Ethical Concerns:**

["NO or VERY MINOR ethics concerns only"]

**Final Justification:**

Thank you for your detailed experiments and response. Your experiments and theoretical analysis have addressed many of my concerns, but regarding MLA’s effectiveness, I believe it must be evaluated from two perspectives. First, MLA only shows an advantage over GQA when training memory is limited. If memory is not constrained—under the same FLOPs or in an overtrained setting—MLA may not offer clear improvement over GQA due to low-rank issues. This is why I emphasize comparing MLA with the same parameter count rather than cache methods, since compute continues to advance and caching may not be the primary scaling direction. Furthermore, once converted, will subsequent training of the model remain unaffected? Is this risk worth taking? In fact, even after MLA’s introduction, most models still train from scratch using GQA. On the other hand, I support the innovation of converting GQA to MLA, but as a KVCache optimization it is too costly and complex, and many alternative methods exist. Moreover, the authors’ design and contributions are only meaningful in the conversion to MLA. However, it remains unclear whether the MLA conversion method will become the mainstream approach for optimizing cache, and therefore its contribution seems limited. Therefore, I have increased my scores for Quality, Clarity, Significance, and Originality.

**Limitations:**

yes

**Quality:**

4

**Strengths And Weaknesses:**

Strengths:
It provides an immediate and feasible path for existing models to utilize DeepSeek's optimization ecosystem, which has strong practical value.

Impressive results were achieved in KV cache compression and inference acceleration. For LLaMA-2-7B, it achieved 93% KV cache compression and 10.6x inference acceleration.

Weakness:
The paper does not make significant improvements to MLA, and the innovations proposed are not the source of major performance improvements (such as kvache compression, speed improvement, parameter pruning, etc.) For example, the paper admits that its "Balance KV technology itself is relatively trivial", but includes it as an "innovation" point as the main contribution.

The method is overly dependent on a specific ecosystem. All the "innovations" in the paper - RoRoPE, FreqFold and even support for Absorb operations - are to achieve compatibility with DeepSeek's unique architecture, but do not discuss the performance differences, advantages and disadvantages of MHA and MLA, and whether there are some improvement solutions. This high degree of coupling has led to doubts about the universality of its methodology, making it impossible to become a general attention mechanism conversion framework with broad scientific value.

---

> ### Author Rebuttal · Authors · 2025-07-31
>
> **Q1. About the innovations proposed in this paper.**
>
> **A1.** This work makes solid theoretical and practical contributions:
>
> **First**, we theoretically prove that MLA has a stronger expressive capacity than GQA under the same KV cache size (see Appendix A), addressing a key gap left by DeepSeek, which only empirically demonstrated MLA’s capabilities. This theoretical framework enables comparative analysis of the expressive power of major attention mechanisms, including MLA, MHA, GQA, and MQA and provides strong motivation for converting GQA to MLA.
>
> **Second**, we are **the first** to prove sufficient conditions for performing invariant query/key transformations across RoPE (see Appendix B), namely, preserving the inner product between RoPE-applied query and key before and after transformation. This breakthrough is crucial for enabling MLA to benefit from inference acceleration **while preserving position information**, which was a significant challenge for prior works. For example, Palu [1] performs low-rank compression before applying RoPE, which not support absorption, requires recomputing the KV representations from the KV cache before computing attention. As a result, the acceleration benefit is limited. MHA2MLA [2] and X-EcoMLA [3] directly truncate RoPE in a non-equivalent manner, leading to significant loss of positional information and degraded performance.
>
> Our theoretical result led to the development of **RoRoPE**, a constrained low-rank compression method, and **FreqFold**, a technique that enhances the expressive power of RoRoPE. Together, these innovations reduce the loss from decoupling RoPE to a minimal level, making GQA-to-MLA conversion with partial RoPE feasible.
>
> Lastly, **Balance-KV** is a simple yet highly effective method for improving joint KV compression. To the best of our knowledge, this technique has not been adopted in prior related works [1,2,3], and we list it as a contribution. Please refer to **Reviewer E7fv A1**, where we provide the theoretical justification behind this method.
>
> **Q2.1. Discuss the performance differences, advantages and disadvantages of MHA and MLA**
>
> **A2.1.** In the *Introduction*, *Related Work*, and the section on *Expressive Power Analysis*, we have outlined the strengths and weaknesses of MHA, GQA, and MLA: 1) **MHA**: High expressive power, but slow inference and maximum KV cache. 2) **GQA**: Lower expressive power, but moderate inference speed. 3) **MLA**: A balance between expressive power and inference efficiency, with higher expressiveness than GQA under the same KV cache. MLA has already garnered significant attention in both model architecture design and AI infrastructure communities. Numerous improved variants such as GLA [4], MTLA [5], and GTA [6] have been proposed, and dedicated frameworks and hardware optimizations for MLA inference are actively under development. Given this momentum, our unified framework for analyzing the expressive capacity of multiple attention mechanisms becomes especially valuable and broadly applicable.
>
> **Q2.2.** The innovations are overly dependent on Absorb operations
>
> **A2.2.** The brilliance of MLA’s design, beyond its use of low-rank compression—a concept already explored in prior works—lies in the **Absorb operation**. This operation is central to MLA’s efficiency: during the **training** phase (which is compute-intensive), it allows attention computations to be performed using compact 192-dimensional representations per head, achieving high computational efficiency. During **inference** (which is memory-bound), it employs a shared 576-dimensional KV cache across heads by absorbing the key projections into the query projection matrices, substantially reducing memory access overhead on both high-bandwidth memory (HBM) and on-chip SRAM.
>
> As demonstrated in Table 3 (rows 2–3), disabling the absorb operation causes MLA’s inference speed at a 16k context length to degrade by **16.7×**, clearly showing its critical role. Consequently, this paper places strong emphasis on supporting the **Absorb operation**, as it is essential for fully realizing MLA’s potential in practical deployment.
>
> The importance of the absorb operation is further illustrated in Palu [1]. In their Figure 5, they show that removing RoPE allows the absorb operation to achieve a **6.17× speedup** at 64K context. However, when RoPE is retained, absorb must be disabled, and the speedup drops to **2.91×**. Unfortunately, Palu does not support RoPE decoupling and therefore cannot apply the absorb operation to popular models such as LLaMA, Qwen, and Mistral.
>
> **Table 3.** Comparison of the impact of num_kv_heads, head_dim, and parameters on inference speed for GQA and MLA. To eliminate the influence of MLP layers on the results, we conducted the speed test using models composed of 32 attention layers only.
>
> | method | num_kv_heads | head_dim | kv_cache per layer | parameters(M) | Throughput(tokens/s) |
> | --- | --- | --- | --- | --- | --- |
> | MLA (with Absorb)  | 1*1 | 576 | 576 | 48.5 | 4151.17 |
> | MLA (without Absorb) | 32*2 | 192 | 12288 | 48.5 | 248.85 |
> | GQA | 4*2 | 128 | 1024 | 37.7 | 2714.80 |
> |  | 8*2 | 128 | 2048 | 41.9 | 1436.92 |
> |  | 16*2 | 64 | 2048 | 41.9 | 1118.83 |
>
> **Q3.1. Why compare capability under equal KV Cache rather than equal parameter count or head dimension?**
>
> **A3.1.** The inference speed of LLMs is primarily affected by two factors: computational intensity and memory access [7]. As the KV cache length increases during inference, modern hardware primarily hits a **memory bottleneck** rather than compute bottleneck. For example, in Table 3, GQA models with the same head dimension where one doubles its KV cache size show drastic change in inference speed (also about **2×**). In contrast, using the same KV cache size but doubling the head dimensions leads to marginal speed changes. For MLA models, they have the same parameter count before and after absorption, but the difference in KV cache size results in a **16.7× gap** in inference speed. Although MLA has more parameters than GQA, its **reduced KV cache size** leads to **faster inference**. Therefore, comparing models under **equal KV cache size**—the factor most directly affecting inference speed—is both practical and meaningful for analyzing expressive capacity.
>
> **Q3.2. There are many ways to handle the KV cache—why focus on MLA?**
>
> **A3.2.** As discussed in *Related Work*, existing KV cache optimization techniques are orthogonal to ours: 1) **KV cache pruning** reduces the number of tokens stored; 2) **KV cache quantization** reduces the bit-width per value. In contrast, TransMLA focuses on **per-token dimensionality reduction**, and thus complements other techniques rather than replacing them.
>
> **Q3.3. Is MLA really better than GQA?**
>
> **A3.3.** Theoretically, we prove that MLA always has greater expressive power than GQA under the same KV cache size. Empirically, several SOTA open-source models using **576-dimensional** MLA—such as **DeepSeek-V3** and **Kimi-K2**—achieve performance comparable to or even surpassing that of GQA models using **2048-dimensional** KV cache, such as LLaMA-4-Maverick-17B-128E, Qwen3-Coder-480B-A35B, Mixtral-8x22B, and Gemma-3-27B. This provides strong evidence that **MLA can effectively replace GQA** as a core component for high-performance, inference-efficient foundation models.
>
> **Q4. Why do GQA models still dominate, and is such a complex conversion to MLA really necessary?**
>
> **A4.** Pretraining a foundation model is a massive system engineering task that requires **months of continuous experimentation and iteration**. Leading model developers have already invested heavily in GQA-based architectures and accumulated significant optimization experience for both training and inference. Switching to a new architecture would demand substantial system reengineering and delays due to tasks like value alignment before public release, not to mention that DeepSeek R1’s release was only 6 months ago. Despite these challenges, we are already seeing a shift—**Kimi-K2**, a state-of-the-art agentic model, has adopted **MLA**. Our theoretical work provides a compelling reason for major model developers to consider this transition. Moreover, the **GQA-to-MLA conversion method** proposed in this paper allows them to **recycle existing pretrained models**, avoiding the need to train from scratch and thereby **reducing the carbon footprint**.
>
> [1] Palu: Compressing KV-Cache with Low-Rank Projection (ICLR 2025)
>
> [2] Towards Economical Inference: Enabling DeepSeek's Multi-Head Latent Attention in Any Transformer-based LLMs (ACL 2025)
>
> [3] X-EcoMLA: Upcycling Pre-Trained Attention into MLA for Efficient and Extreme KV Compression (COLM 2025)
>
> [4] Hardware-Efficient Attention for Fast Decoding
>
> [5] Multi-head Temporal Latent Attention
>
> [6] Grouped-head latenT Attention
>
> [7] Roofline: an insightful visual performance model for multicore architectures.

---

> ### Author Response · Authors · 2025-08-05
> **Response to Reviewer i2PF**
>
> Thank you for supporting the innovation of converting GQA to MLA, and for recognizing our paper’s Quality, Clarity, Significance, and Originality. In response to your new question, we reply as follows:
>
> **Q1. TransMLA is too costly and complex, and many alternative KV-cache optimization methods exist.**
>
> **A1.** The rigorous theoretical guarantees in TransMLA (including the expressivity proof and the proof of rotation invariance under RoPE) may give the impression that the method is costly and complex. Setting those proofs aside, TransMLA ultimately just applies PCA twice—first to decouple RoRoPE, and second to compress the KV cache. The PCA is widely used across various KV-cache compression methods (Palu, ICLR 2025; RazorAttention, ICLR 2025; MatryoshkaKV, ICLR 2025; MHA2MLA, ACL 2025; CLOVER, ICML 2025; Loki, NeurIPS 2024). Therefore, TransMLA itself is not costly or complex. Moreover, as we have already noted, other KV-cache compression techniques such as sparse attention and KV-cache quantization are compatible with TransMLA.
>
> **Q2. MLA only helps when training memory is limited. If memory is unconstrained—at equal FLOPs or in an overtrained regime—MLA may not clearly outperform GQA due to low-rank issues.**
>
> **A2.** Pretraining LLMs is extremely expensive. For example: DeepSeek V3 pretraining on 2,788K H800 GPU-hours cost 5.576M dollars, and Llama 3.1 405B pretraining on 30.84M H100 GPU-hours cost 61.68M dollars. Therefore, assuming unconstrained training memory is meaningless. Tri Dao, the author of FlashAttention and Mamba, points out: “LLM decoding is bottlenecked for large batches and long contexts by loading the key-value (KV) cache from high-
> bandwidth memory, which inflates per-token latency, while the sequential nature of decoding limits parallelism.” [1]
>
> The number of parameters does not directly determine a model’s speed; it influences speed indirectly through floating-point operations (FLOPs) and memory operations. However, “Modern GPUs devote considerably more silicon to computation than memory bandwidth, a trend that intensified in 2017. Hardware FLOPs have scaled by∼3×every two years, while the HBM memory bandwidth increases by∼1.6×over the same period.” [1]
>
> According to the roofline model [2], Arithmetic Intensity = FLOPs / memory operations. Take the NVIDIA H800 SXM5 GPU as an example: it has a peak memory bandwidth of 3.35 TB/s and a practical peak of 865 TFLOPs. When Arithmetic Intensity > 256, the kernel is compute-bound; otherwise, it is memory-bound.
> For GQA,
> $\text{Arithmetic Intensity}=\frac{2 h_q s_{kv} d}{2 h_q d + 2 h_{kv} s_{kv} d}\approx \frac{h_q}{k_{kv}}$.
> For LLaMA-3-70B, $\frac{h_q}{k_{kv}} = \frac{64}{8} = 8$. Thus it is memory-bound, meaning the data fed into the GPU cannot keep up with the GPU’s processing speed, resulting in low GPU utilization.
> For MLA,
> $\text{Arithmetic Intensity} = h_q \cdot \frac{d_k + d_v}{d_k} \approx 2 h_q$.
> For DeepSeek V3, $2 h_q = 256$, which balances compute and memory throughput and therefore lets the GPU achieve high utilization.
>
> This is the theoretical basis for why we emphasize the importance of the KV cache.
>
> **Q3. The authors’ design and contributions are only meaningful in the conversion to MLA. After MLA’s introduction, most models still train from scratch using GQA.**
>
> **A3.** Which of GQA or MLA will ultimately dominate large-scale model design remains an open question. This paper argues—on theoretical grounds—that MLA has an expressivity advantage over GQA. Intuitively, each attention head needs to retain some repeated information; under GQA, every group must keep its own copy of that information, whereas under MLA all heads share a single KV cache. Consequently, to preserve the same information, MLA requires a lower rank. Our aim is to draw broader attention to MLA and, additionally, to provide a low-cost pathway for migrating existing models to MLA.
>
> Thank you again for your thoughtful feedback; we would be pleased to continue the discussion as needed. If our updated responses address your concerns, we would be grateful if you could consider raising your score.
>
> [1] Hardware-Efficient Attention for Fast Decoding
>
> [2] Roofline: an insightful visual performance model for multicore architectures.

---

> ### Author Response · Authors · 2025-08-05
> **Supplementary Explanation**
>
> Dear Reviewer,
>
> We recognize that the transition from GQA to MLA may feel strange to you—here’s one way to understand about it. There are many techniques for **compressing the dimensionality** of the KV cache that are orthogonal to (i) reducing the number of tokens and (ii) lowering numerical precision via quantization. **Building on these dimensionality-compression methods, we further enable MLA’s matrix-absorption capability**. This design allows TransMLA to reuse FlashMLA’s inference-optimized kernels, enabling not only easier and faster deployment across DeepSeek-compatible environments compared to approaches that require custom kernels, but also delivering better acceleration performance.

---

> > ### Author Response · Authors · 2025-08-07
> > **Concern about your overall score**
> >
> > Dear Reviewer i2PF,
> >
> > Has our reply completely resolved your concerns? The reviewer-author discussion phase is about to end, do you believe the innovations in our paper and efforts during the rebuttal deserve acceptance as a conference paper, and provides the community with:
> > 1. a method for equivariant rotation execution beyond RoPE,)
> > 2. a perspective comparing MLA and GQA,
> > 3. and a method using MLA’s absorb operation to enhance inference acceleration?
> > We hope to earn your recognition, and will continue advancing the TransMLA project, continually enhancing its impact and practical value.
> >
> > best regard，
> >
> > Authors

---

> ### Author Response · Authors · 2025-08-09
> **Thank you for raising your score**
>
> Dear Reviewer i2PF,
>
> Thank you very much for your thoughtful feedback and for raising your score. We sincerely appreciate your input, which has been invaluable in improving our work.
>
> We will ensure that all of your suggestions are carefully considered and incorporated into the camera-ready version of the paper.
>
> Once again, thank you for your support and encouragement.
>
> Best regards

---

### Official Review · Reviewer_qHce · 2025-07-03

**Clarity:** 3
**Significance:** 3
**Originality:** 3
**Rating:** 4
**Confidence:** 2

**Summary:**

The authors propose a comprehensive method for converting models based on MHA, as well as simple variants such as GQA, into MLA-compatible architectures, aiming to achieve enhanced expressiveness and overall efficiency. The proposed approach first performs Key fusion, followed by tailored handling of RoPE-related Query and Key components to better adapt them to the specialized RoPE mechanisms in MLA, namely RoRoPE and FreqFold. Finally, a joint PCA-based dimensionality reduction is applied to the Key and Value matrices, guided by the distributional characteristics of their weights. The result is a model that can be effectively adapted to MLA. The authors validate their approach by conducting conversion experiments on models of two different scales, followed by fine-tuning on a small amount of tokens, demonstrating the effectiveness of their method in restoring model capabilities. Furthermore, ablation studies are provided to underscore the necessity of the proposed RoPE handling and weight-based dimensionality reduction strategies.

**Questions:**

Could the authors provide updated results using more recent models, and align the recovery training data with the original pretraining data to enhance the credibility and relevance of the experimental section?

**Ethical Concerns:**

["NO or VERY MINOR ethics concerns only"]

**Final Justification:**

The authors have addressed my concern regarding the limited scope of model testing. However, the issue of insufficient adaptability to GQA models remains. Furthermore, the work is oriented more toward theoretical exploration rather than the proposal of an engineering-driven method. Nevertheless, I believe this manuscript carries sufficient significance and value, and its level of completeness, particularly after the addition of the supplementary experiments, is satisfactory.

**Limitations:**

yes

**Paper Formatting Concerns:**

No major concerns.

**Quality:**

3

**Strengths And Weaknesses:**

**Strengths**

1. The authors design an effective training pipeline that enables the transformation of non-native MLA models into their MLA-compatible counterparts. The empirical results validate the method's effectiveness. In particular, the RoPE adaptation and KV compression techniques offer valuable insights for researchers working on MLA conversion and optimization.

2. The manuscript presents results across different compression scales. Notably, at lower compression ratios, TransMLA is able to retain strong performance even without additional training.

3. The authors employ well-designed visualizations and empirical studies to convincingly demonstrate the necessity and effectiveness of their algorithmic components.

**Weaknesses**

1. A major limitation of the manuscript lies in the choice of baseline models. Both SmolLM-1.7B and LLaMA-2-7B are based on MHA, which, according to prior work and empirical observations, tends to be particularly amenable to KV compression. In contrast, more recent and mainstream models often adopt GQA or similar attention variants, which are known to present greater challenges for compression, especially in the KV cache. As such, the experimental results presented may have limited relevance to current model architectures.

2. The capability restoration experiment for LLaMA-2-7B uses the same training data as that used for SmolLM-1.7B. However, LLaMA-2-7B was trained in the early 2023 using a different dataset. The data used for post-conversion fine-tuning may thus be of higher quality than the original pretraining corpus, implying that the observed gains could result not from genuine restoration, but from additional learning. This undermines the interpretability of the experimental results as a strict recovery study.

---

> ### Author Rebuttal · Authors · 2025-07-31
>
> Thank you for the insightful questions.  Below we provide point-by-point responses to your comments. All suggested improvements will be incorporated into the camera-ready version.
>
> **Q1. The experiments focus on MHA models, but most current models use GQA, which is harder to compress.**
>
> **A1.** As stated in lines 29–36 of the original paper, both MHA and MQA can be seen as special cases of GQA. Specifically, when the number of query heads equals the number of key-value heads in GQA, it effectively becomes an MHA. Our goal is to compare TransMLA with prior relevant work under consistent and fair experimental settings. To this end, we selected the recently published MHA2MLA [1] as our baseline, as it also targets RoPE decoupling and KV cache compression. Following their experimental setup, we used two MHA-based models—SmolLM-1.7B and LLaMA-2-7B. Moreover, **TransMLA is fully compatible with GQA**, all of our ablation studies were performed on LLaMA-3-8B, a GQA-based model.
>
> Nevertheless, as you suggested, we extended our evaluation to several mainstream GQA models. The results are reported in Table 1.
>
> **Table 1.** Evaluated with Wikitext-2 Perplexity (PPL) of Converting GQA Models to MLA (lower is better). Original PPL refers to the perplexity of the original GQA model without conversion. Decoupled RoPE PPL represents the result after applying RoRoPE and FreqFold to reduce decoupled RoPE loss. Compressed KV PPL indicates the perplexity when the KV cache is compressed to 576 with the help of Balance-KV.
>
> | Model | Original ppl | Decouple RoPE ppl | Compress KV  ppl | KV Cache |
> | --- | --- | --- | --- | --- |
> | Llama-3-8B | 6.1371 | 8.3997 | 18.35 | -71.875% |
> | Qwen2.5-7B | 6.848 | 7.3059 | 7.9812 | -43.75% |
> | Qwen2.5-72B | 4.2687 | 4.6931 | 7.7172 | -71.875% |
> | Mistral-7B | 5.3178 | 5.5915 | 7.0251 | -71.875% |
> | Mixtral-8x7B | 3.8422 | 4.1407 | 5.8374 | -71.875% |
>
> From Table 1, we can observe that for GQA models—ranging from 7B to 70B, whether dense models or MoE models—TransMLA consistently incurs only minimal performance loss. It’s worth noting that MLA is specifically optimized for a latent dimension of 576, which we adopt as a unified compression target across all models. Since GQA models typically maintain smaller KV caches than MHA, converting them to MLA results in less information loss, making the transition both effective and efficient. Due to time constraints, we only present the results of the conversion without additional training. We plan to include the full alignment performance for all models in the camera-ready version.
>
> **Q2. The restoration experiment for LLaMA-2-7B uses different data from its original training. Could the improvements come from additional learning rather than true recovery?**
>
> **A2.** As mentioned in A1, our work follows the experimental setup of MHA2MLA and uses the smollm-corpus for pretraining. We noted that the pretraining corpus for LLaMA-2-7B has not been publicly released. To address this concern, we adopted the data distillation approach proposed in the LLM-QAT [2] paper, generating a new dataset from the original model to guide the distillation of the converted model. For each model, we distilled datasets containing 14,000 samples, with each sample consisting of 2,048 tokens. These datasets were used to distill the converted model by minimizing the error between the output of each converted MLA attention module and the original model's corresponding GQA outputs, as well as aligning with the final token prediction. Each attention module was trained for 10 epochs, and we perform hyperparameter tuning over 10 evenly spaced learning rates in the range [1e-3, 2e-2]. The results are presented in Table 2.
>
> **Table 2.** Commonsense reasoning accuracy for distilling converted MLA-based models from original GQA models
>
> | Model | KV cache | avg | arc | hellaswag | mmlu | openbook_qa | piqa | winogrande |
> | --- | --- | --- | --- | --- | --- | --- | --- | --- |
> | LLaMA-2-7b | — | 0.5981 | 0.5931 | 0.7313 | 0.4135 | 0.4160 | 0.7840 | 0.6504 |
> | TransMLA |  -68.75% | 0.5989 | 0.5982 | 0.7243 | 0.4167 | 0.4260 | 0.7818 | 0.6464 |
> | Qwen2.5-7b | — | 0.6338 | 0.6445 | 0.7749 | 0.4726 | 0.4540 | 0.7976 | 0.6590 |
> | TransMLA | -71.875% | 0.6312 | 0.6813 | 0.7490 | 0.4633 | 0.4420 | 0.7911 | 0.6606 |
> | Mistral-7b-v0.1 | — | 0.6483 | 0.6671 | 0.7948 | 0.4567 | 0.4560 | 0.8286 | 0.6867 |
> | TransMLA | -50% | 0.6448 | 0.6622 | 0.7900 | 0.4553 | 0.4560 | 0.8270 | 0.6780 |
>
> According to Table 2, aligning the converted models using only the distilled data from the original models allows them to closely match the performance of the original models. This confirms that the observed performance improvements are not due to additional learning but reflect genuine restoration. We plan to distill more data and complete experiments on additional models and compression rates in the future.
>
> [1] Towards Economical Inference: Enabling DeepSeek's Multi-Head Latent Attention in Any Transformer-based LLMs (ACL 2025)
>
> [2] LLM-QAT: Data-Free Quantization Aware Training for Large Language Models (ACL Findings 2024)

---

> ### Comment · Reviewer_qHce · 2025-08-06
>
> I would like to reiterate that my concern is not with the fundamental incompatibility between GQA and TransMLA, but rather with the trade-off between compression rate and final performance. That said, the authors have provided key results on the main GQA-based model, which I believe sufficiently address the logical gap previously present in the manuscript. I will revise my score accordingly, pending further consideration of the appropriate adjustment. I wish the authors all the best.

---

> > ### Author Response · Authors · 2025-08-06
> > **Thank you for raising your score.**
> >
> > We're very pleased that you found our experimental results address your concerns. Regarding the relationship between compression rate and performance, our empirical observation is: Models with inherently larger KV cache tend to have more redundancy. For these, a significant portion of the cache (e.g., 68.75% in MHA) can often be reduced with minimal performance degradation (only ~1.6% loss) even without additional training. However, for models with less inherent redundancy, such as GQA, or when pushing compression rates even higher in architectures like MHA, the impact on performance becomes noticeably more pronounced. In these cases, considerably more distillation training is required to recover the model's effectiveness.
> >
> > We will conduct more comprehensive ablation studies analyzing the compression rate vs. performance trade-off in the camera-ready version of the manuscript.
> >
> > Finally, thank you once again for your willingness to raise your score. This is tremendous encouragement for us.

---

### Author Response · Authors · 2025-08-04

**Dear Reviewers,**

We would like to express our sincere gratitude for the valuable feedback provided by each of you. Below, we summarize the key contributions of our original submission and the additional experiments we conducted during the rebuttal phase to address the concerns raised. We hope that our efforts have sufficiently addressed your questions.

### Summary of Our Original Submission:

1. **Overall Contribution**: We proof that MLA outperforms GQA in terms of expressiveness under identical KV cache conditions. Additionally, we introduced a method called TransMLA, which converts existing MHA/GQA models into MLA models. This conversion process allows us to directly reuse optimized kernels for MLA, such as those in FlashMLA, FlashAttention, and Triton, without the need to develop new CUDA kernels, enabling immediate inference acceleration.

2. **Orthogonal Transformations across RoPE**: For the first time, we proved that equivalent orthogonal transformations across RoPE are feasible, provided that:
   - Rotations occur within the same dimension of each attention head, and
   - Both the real and imaginary parts of RoPE use the same rotation scheme.

3. **RoRoPE and FreqFold Methods**: Based on the theoretical insights, we proposed the *RoRoPE* and *FreqFold* methods, which significantly mitigate positional information loss caused by decoupling RoPE. These methods ensure that the converted MLA can support the absorb operation.

4. **Balance KV Norm Trick**: We introduced the *Balance KV norm trick*, which notably reduces the loss associated with joint compression of the KV cache.

### Additional Experiments During the Rebuttal Phase:

To further validate TransMLA and address the reviewers' concerns, we conducted extensive additional experiments and provided detailed explanations. These included:

1. **Model Support**: We tested TransMLA on various models, including Llama-3-8B, Qwen2.5-7B, Qwen2.5-72B, Mistral-7B, and Mixtral-8x7B, confirming that TransMLA supports a wide range of MHA and GQA models.

2. **Distilled MHA Models**: We demonstrated that TransMLA can restore model capabilities using distilled MHA model data without introducing additional information.

3. **Speed Improvements**: We empirically validated the significant speed improvements achieved through the absorb operation, highlighting the importance of RoRoPE and FreqFold for TransMLA.

4. **Proof for the alignment of the KV cache**: We performed comparative experiments on parameter count, head dimensions, and KV cache effects on speed, justifying the alignment of the KV cache as a basis for comparing model expressiveness.

5. **Ablation Studies**: We conducted ablation studies on calibration data sources and dataset sizes to assess their impact on performance.

6. **Effect of Innovations**: Ablation experiments on RoRoPE, FreqFold, and BalanceKV demonstrated that each of these innovations contributes significantly to improving the final conversion effect.

7. **LongBench Benchmark**: Our experiments on the LongBench benchmark for long-text tasks showed that TransMLA maintains excellent performance on long-text inference tasks.

8. **Acceleration on Hopper GPUs**: We validated TransMLA’s 11x acceleration in long-text inference using FlashMLA on Hopper GPUs.

9. **Theoretical Foundation of Balance KV Norm**: We provided a theoretical proof that the *Balance KV norm PCA* minimizes reconstruction error, reinforcing the theoretical justification for this trick.

10. **KV Cache Compression**: Additional ablation experiments demonstrated the effectiveness of the Balance KV factor in KV cache compression.

### Reviewer-Author Discussion Update:

As the Reviewer-Author Discussion period is now halfway through, we are eager to continue addressing any additional questions or concerns you may have. We believe that TransMLA offers robust theoretical and methodological innovations, establishing a quantitative benchmark for the expressiveness of new attention mechanisms. It presents a compelling motivation and practical approach for transitioning large models from GQA to MLA, facilitating the reuse of existing GQA models and enabling their direct transformation into faster and more capable MLAs.

We sincerely appreciate your attention to this matter and look forward to hearing whether our rebuttal has sufficiently addressed your concerns, or if you are willing to revise your scores.

Best regards,
**Submission 3063 Authors**

---

### Author Response · Authors · 2025-08-09

Dear All Reviewers,

Thank you for your time and expertise in reviewing our work. As the reviewer-author discussion period concludes shortly, we kindly remind you to confirm your final score and submit the Mandatory Acknowledgement

We sincerely appreciate your insightful feedback and suggestions. We will carefully incorporate your recommendations into the camera-ready version of the paper and actively contribute to the community by open-sourcing all related resources, including:

• Transformation, training, and inference code

• Model weights

This will ensure full reproducibility and transparency.

Best regards,

The Authors

---

### Note · Authors · 2025-08-13

Dear All,

The original TransMLA paper has two core contributions:

1.	Demonstrating that MLA has greater expressive power than GQA under same KV Cache.

2.	Converting GQA to MLA — Using RoRoPE, FreqFold, BalanceKV achieving up to 10× speedup by directly using Deepseek’s inference code.

**During the rebuttal period**, we

- added experiments on multiple GQA models.

- Trained with distilled data and used speed tests to demonstrate the importance of the absorb operation.

- Showed that the KV cache has the greatest impact on speed, and that TransMLA is orthogonal to other KV cache compression methods.

- Conducted ablation studies on calibration, as well as on RoRoPE, FreqFold, and BalanceKV.

- Evaluated on the LongBench benchmark and tested FlashMLA on Hopper GPUs.

- Provided a theoretical foundation for Balance KV Norm and analyzed BKV’s impact on KV cache compression.

**During the Author–Reviewer discussion period:**

- **Reviewer qHce** acknowledged that the new results on GQA-based models address the previously identified logical gap and **stated they will revise their score**.

- **Reviewer i2PF** requested discussion on the case where memory is not constrained — we argued that this setting is unrealistic. The reviewer noted that many alternative KV cache optimization methods exist. We clarified that these methods are orthogonal to TransMLA and can be combined. When questioned whether TransMLA is only meaningful for conversion to MLA, we highlighted MLA’s importance.
They  expressed concern over performance degradation, and we explained that RoRoPE, FreqFold, and BalanceKV were specifically designed to minimize this drop. They  **improved their score**.

- **Reviewer KGKd** asked how TransMLA handles QKV bias — we explained our approach. They were concerned that FlashMLA only supports 576 dimensions — we showed that grouping can support dimensions beyond 576 and provided experimental results showing its effectiveness and speed. They noted that MLA has more parameters — we confirmed this and pointed out that it is accounted for in our proofs. **KGKd stated they will seriously consider raising their score.**

At this point, we have **addressed all reviewers’ concerns** and will incorporate their suggestions from the rebuttal into the camera-ready version. We sincerely **thank Reviewers qHce, i2PF, and KGKd for their willingness to raise their scores**, and **Reviewer E7fv for recognizing TransMLA’s contributions and awarding a score of 5**.

---

### Decision · Program_Chairs · 2025-09-17

**Decision:**

Accept (spotlight)

**Comment:**

This paper proposes TransMLA, a practical framework for converting MHA/GQA models into MLA-compatible ones, enabling significant KV cache compression and inference acceleration. Key innovations like RoRoPE, FreqFold, and Balanced KV PCA are well-motivated and validated. The paper stands out in terms of empirical rigor and practical relevance. While scope limitations (e.g., lack of GQA baselines, limited theoretical grounding) were noted, the authors addressed most concerns in rebuttal with additional experiments and clarifications. This work is an accept.